# Frequency Adaptive Normalization For Non-stationary Time Series Forecasting

**Weiwei Ye**[1]    **Songgaojun Deng**[2]    **Qiaosha Zou**[3]    **Ning Gui**[1]*

[1]Central South University    [2]University of Amsterdam    [3]Zhejiang Lab

wwye155@gmail.com, s.deng@uva.nl, qiaoshazou@zhejianglab.org, ninggui@gmail.com

## Abstract

Time series forecasting typically needs to address non-stationary data with evolving trend and seasonal patterns. To address the non-stationarity, reversible instance normalization has been recently proposed to alleviate impacts from the trend with certain statistical measures, e.g., mean and variance. Although they demonstrate improved predictive accuracy, they are limited to expressing basic trends and are incapable of handling seasonal patterns. To address this limitation, this paper proposes a new instance normalization solution, called frequency adaptive normalization (FAN), which extends instance normalization in handling both dynamic trend and seasonal patterns. Specifically, we employ the Fourier transform to identify instance-wise predominant frequent components that cover most non-stationary factors. Furthermore, the discrepancy of those frequency components between inputs and outputs is explicitly modeled as a prediction task with a simple MLP model. FAN is a model-agnostic method that can be applied to arbitrary predictive backbones. We instantiate FAN on four widely used forecasting models as the backbone and evaluate their prediction performance improvements on eight benchmark datasets. FAN demonstrates significant performance advancement, achieving 7.76%~37.90% average improvements in MSE. Our code is publicly available[2].

## 1 Introduction

Time series forecasting plays a key role in various fields such as traffic [8], finance [23] and infectious disease [1], etc. Recent research focuses on deep learning-based methods, as they demonstrate promising capabilities to capture complex dependencies between variables [47, 42, 46]. However, time series with trends and seasonality, also called non-stationary time series [15], create covariate pattern shifts across different time steps. These dynamics pose significant challenges in forecasting.

To mitigate non-stationarity issues, reversible normalization has been proposed [33, 17] which first removes non-stationary information from the input and returns the information back to rebuild the output. Current work focuses on removing non-stationary signals with statistical measures, e.g., mean and variance in the time domain [10, 28]. However, while these methods have improved prediction accuracy, these statistical measures are only capable of extracting the most prominent component, i.e., the trend, leaving substantial room for improvement. They, we argued, can hardly measure the characteristics of seasonal patterns, which are quite common in many time series. This significantly limits their capability to handle the non-stationarity, especially the seasonal patterns.

We illustrate an example featuring one of the simplest non-stationary signals in Fig. 1. This graph shows a time-variant signal with a gradually damping frequency, which is widely seen in many passive systems, e.g., spring-mass damper systems. As depicted in Fig. 1, the three input stages

---

*Corresponding author

[2]http://github.com/wayne155/FAN

(highlighted in different background colors) exhibit the same mean and variance but differ in Fourier frequencies. Previous methods that model non-stationary information using means and variances can hardly distinguish this type of change in the time domain. In comparison, changes in periodic signals can be easily identified with the instance-wise Fourier transform ($f_1 \neq f_2 \neq f_3$). Thus, in this context, the principal Fourier components provide a more effective representation of non-stationarity compared to statistical values such as mean and variance. This simple example also shows that many existing frequency-based solutions, e.g., TimesNet [41], Koopa [27], which assume that the principal frequencies of the input signal are constant, can not identify the evolving principal frequencies.

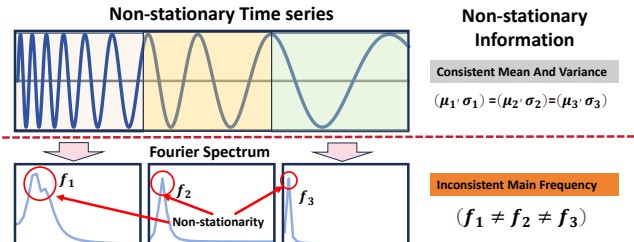

Figure 1: A sinusoidal signal with linearly varying frequency which is a common example of a non-stationary time series. In the lower-left corner, we plot the Fourier spectrum for three segments of the signal.

With this inspiration, we introduce a novel instance-based normalization method, named **F**requency **A**daptive **N**ormalization (FAN). Rather than normalizing temporal statistical measures, FAN mitigates the impacts from the non-stationarity by filtering top $K$ dominant components in the Fourier domain for each input instance, this approach can handle unified non-stationary fact composed of both trend and seasonal patterns. Furthermore, as those removed patterns might evolve from inputs to outputs, we employ a pattern adaptation module to forecast future non-stationary information rather than assuming these patterns remain unchanged.

In summary, our main contributions are: 1) We illustrate the limitations of reversible instance normalization methods in using temporal distribution statistics to remove impacts from non-stationarity. To address this limitation, we introduce a novel reversible normalization method, named FAN, which adeptly addresses both trend and seasonal non-stationary patterns within time series data. 2) We explicitly address pattern evolution with a simple MLP that predicts the top $K$ frequency signals of the horizon series and applies these predictions to reconstruct the output. 3) We apply FAN to four general backbones for time series forecasting across eight real-world popular benchmarks. The results demonstrate that FAN significantly improves their predictive effectiveness. Furthermore, a comparative analysis between FAN and state-of-the-art normalization techniques underscores the superiority of our proposed solution.

## 2 Related Work

Time series forecasting has been a hot topic of study for many years. This section provides discussions on related work from the following three perspectives.

### 2.1 Time Series Forecasting

Traditional statistical methods, such as ARIMA [2], assume the stationarity of the time series and dependencies between temporal steps. Although these methods provide theoretical guarantees, they typically require data with ideal properties, which is often inconsistent with real-world scenarios [42]. Besides, they can only handle a limited amount of data and features. In recent years, the field has witnessed a significant proliferation in the application of deep learning techniques for multivariate time series forecasting, a development ascribed to their ability in handling high-dimensional datasets. Consequently, various methods have been proposed to model time series data. Work based on recurrent neural networks [36, 4] preserves the current state and models the evolution of time series as a recurrent process. However, they generally suffer from a limited receptive field, which restricts their ability to capture long temporal dependencies [47]. Inspired by their successes in Computer Vision (CV) and Natural Language Processing (NLP), convolutional neural networks and the self-attention mechanism have been extensively utilized in time series forecasting [22, 19, 40, 25]. Nevertheless, those works still face difficulties in handling non-stationary data with covariate pattern shifts. Making an accurate prediction for non-stationary time series remains challenging.

## 2.2 Non-stationary Time Series Forecasting

To address non-stationarity, many methods directly model non-stationary phenomena with different modeling techniques. Li et al. [24] utilize a domain-adaptation paradigm to predict data distributions. Du et al. [6] propose an adaptive RNN to alleviate the impact of non-stationary factors through distribution matching. Liu et al. [26] introduce a non-stationary Transformer with de-stationary attention that incorporates non-stationary factors in self-attention mechanisms. To model non-linear dynamic systems, several models based on Koopman theory [21, 29, 37, 44] have been proposed with Fourier transform. To learn different patterns at different scales, Wang et al. [38] employs global and local Koopman operators. Liu et al. [27] model non-stationarity identified with Fourier transform and use Koopman operators to learn those components. However, these solutions typically select fixed frequency components based on the whole sequence rather than frequencies based on inputs. This time-invariant assumption can hardly be true in real-world scenarios.

## 2.3 Instance-wise Normalization against Non-stationarity

To mitigate the time-variant property of non-stationary time series, a set of instance-wise normalization methods have been proposed to remove the impacts from non-stationarity. To reflect instance-wise changes, Ogasawara et al. [31] propose the usage of normalization based on local properties rather than global statistics. Passalis et al. [33] introduce an adaptive and learnable approach to this instance-wise normalization paradigm. However, although these methods effectively remove non-stationary components from inputs, they still need to predict the non-stationary time series in the output series, which remains challenging. In response, reversible instance normalization [17] is introduced by reintegrating the removed non-stationary components back to reconstruct the output. However, it still assumes unchanged trends between inputs and outputs. Kim et al. [17] developed RevIN, which mainly addresses evolving trends between input sequences. Recent works [10, 28] explore trends at a finer granularity, e.g., at the sliced level. However, these approaches still model non-stationarity with temporal statistical distribution parameters and fail to account for evolving seasonality, which is a crucial aspect of non-stationarity [35, 11, 45].

## 3 Proposed Method: FAN

Given a multivariate time series $\mathcal{X} \in \mathbb{R}^{N \times D}$, where $N$ is the total time steps and $D$ denotes the number of feature dimensions. We use inputs series with length $L$ to predict outputs series within length $H$. The forecast task can be formulated as: $\mathcal{X}_{t-L:t} \to \mathcal{X}_{t+1:t+H}$, where $\mathcal{X}_{t-L:t} \in \mathbb{R}^{L \times D}$ and $\mathcal{X}_{t+1:t+H} \in \mathbb{R}^{H \times D}$. For a clearer notation, we denote the input and output series as $\mathbf{X}_t \in \mathbb{R}^{L \times D}$ and $\mathbf{Y}_t \in \mathbb{R}^{H \times D}$ respectively.

Our proposed method, FAN, consists of symmetrically structured instance-wise normalization and denormalization layers, illustrated in Fig. 2. The normalization process removes the impacts of non-stationary signals through frequency domain decomposition (upper left part of Fig. 2), while the denormalization process, supported by a prediction module, addresses potential shifts in frequency components between the input and output (lower part of Fig. 2).

### 3.1 Frequency-based Normalization

First, FAN removes the top $K$ dominant components in the frequency domain for each input instance, so the forecasting backbone can concentrate on the stationary aspects of the input. We term this process as *Frequency Residual Learning* (FRL). The input at time $t$, $\mathbf{X}_t$, is multivariate with $D$ dimension, and each dimension might have different frequency patterns; thus, we apply the FRL to each dimension in a channel independence setting [30]. Here, the FRL is realized by the 1-dim Discrete Fourier Transform (DFT) with $\mathrm{DFT}(\cdot)$ towards each input $\mathbf{X}_t$:

$$\mathbf{Z}_t = \mathrm{DFT}(\mathbf{X}_t) \quad \text{and} \quad \mathcal{K}_t = \mathrm{TopK}(\mathrm{Amp}(\mathbf{Z}_t)) \quad \text{and} \quad \mathbf{X}_t^{non} = \mathrm{IDFT}(\mathrm{Filter}(\mathcal{K}_t, \mathbf{Z}_t)) \tag{1}$$

Equ. 1 shows that $\mathrm{DFT}(\cdot)$ transforms an input into Fourier components in complex values, denoted $\mathbf{Z}_t \in \mathbb{C}^{T \times D}$. Then, $\mathrm{TopK}(\cdot)$ selects the frequency set with the top $K$ largest amplitude, which are calculated with $\mathrm{Amp}(\cdot)$ function. $\mathrm{Filter}$ is the operation to filter out the $\mathcal{K}_t$ frequency from $\mathbf{Z}_t$. To mitigate the impact of non-stationary signals, FRL restores the top $K$ components into time domain components $\mathbf{X}_t^{non}$ with $\mathrm{IDFT}(\cdot)$.

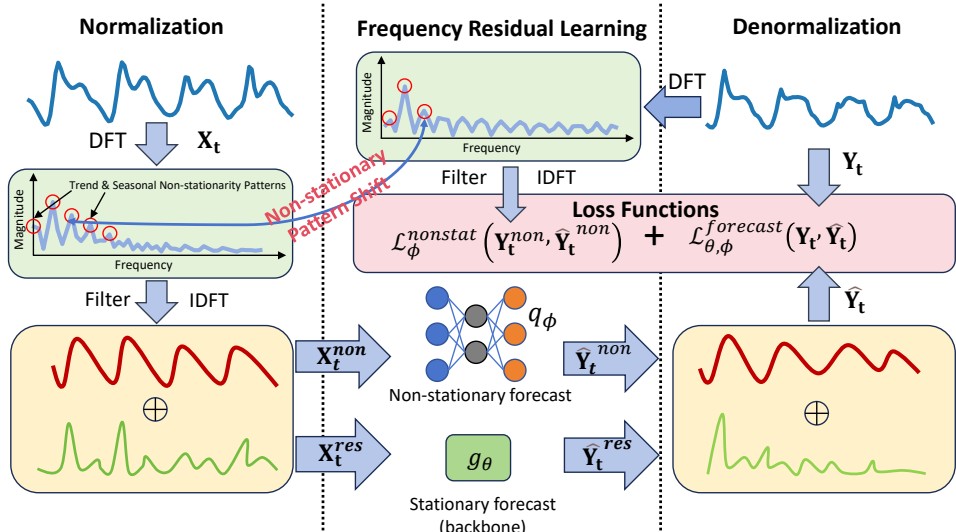

Figure 2: An overview of FAN which consists of normalization, frequency residual learning, denormalization steps, and incorporates a prior loss for non-stationary patterns.

With $\mathbf{X}_t^{non}$, we can easily normalize the inputs and get the stationary components by removing $\mathbf{X}_t^{non}$ from $\mathbf{X}_t$, which is

$$\mathbf{X}_t^{res} = \mathbf{X}_t - \mathbf{X}_t^{non} \tag{2}$$

Here, $\mathrm{DFT}(\cdot)$ and $\mathrm{IDFT}(\cdot)$ can be performed using Fast Fourier Transform (FFT) [3] with a computational complexity of $O(L \log L)$. And the $\mathrm{TopK}$ and $\mathrm{Filter}$ operations both exhibit complexity of $O(L + K)$. It is important to note that all these operations are GPU-friendly and can be fully paralleled. Thus, the impact of applying these operations independently on each dimension can be largely mitigated. GPU-friendly PyTorch pseudocode is in Appendix A.2. After the normalization step, the normalized sequences $\mathbf{X}_t^{res}$ can be more stationary and have a more consistent covariate distribution, the theoretical proof is provided in Appendix C.

### 3.2 Forecast & Denormalization

As a result, the normalization layer allows the forecast backbone model $g_\theta$ to focus more on the dynamics within the inputs. Here, following the reversible instance normalization paradigm, the forecast backbone $g_\theta$ receives the transformed data $\mathbf{X}_t^{res}$ as input and forecasts only the stationary part $\mathbf{Y}_t^{res}$ of the outputs $\mathbf{Y}_t$. This design makes it easier for the model to forecast non-stationary time series. Then, we apply the removed non-stationary information back to the output. We define this process as:

$$\begin{aligned} \hat{\mathbf{Y}}_t^{res} &= g_\theta(\mathbf{X}_t^{res}) \\ \hat{\mathbf{Y}}_t &= \hat{\mathbf{Y}}_t^{res} + \hat{\mathbf{Y}}_t^{non} \end{aligned} \tag{3}$$

where $\hat{\mathbf{Y}}_t^{non}$ is the reconstruct signal of $\mathbf{X}_t^{non}$. We illustrate $\hat{\mathbf{Y}}^{non}$ as follow:

**Non-stationary shift forecasting.** For reverse instance normalization, we need to estimate $\hat{\mathbf{Y}}_t^{non}$ in the outputs. As an input and its corresponding output are rather close, RevIN [17] directly adds $\mathbf{X}_t^{non}$ back by assuming $\mathbf{Y}_t^{non}$ with exactly the same trend as $\mathbf{X}_t^{non}$. However, this assumption can hardly be true as the non-stationary information between the input and output may evolve. Furthermore, although SAN [28] and Dish-TS [10] predict statistics to address the discrepancy between the input and output, these statistics can only represent the most salient trend patterns.

To this end, rather than predicting statistics [10, 28], we use a simple MLP model $q_\phi$ to directly predict future values of the composite top $K$ frequency components for $D$ dimensions, defined as:

$$\hat{\mathbf{Y}}_t^{non} = q_\phi(\mathbf{X}_t^{non}, \mathbf{X}_t) = \mathbf{W}_3 \, \mathrm{ReLU} \left( \mathbf{W}_2 \, \mathrm{Concat}(\mathrm{ReLU} \left( \mathbf{W}_1 \mathbf{X}_t^{non} \right), \mathbf{X}_t) \right) \tag{4}$$

where $\mathbf{W}_1, \mathbf{W}_2, \mathbf{W}_3$ are all learnable parameters. Here, since $\mathbf{X}_t^{non}$ only contains top $K$ frequency information, it is difficult to capture variations in other frequencies solely relying on $\mathbf{X}_t^{non}$. Therefore,

we concatenate the top $K$ components with the original input $\mathbf{X}_t$ to handle potential frequency variations.

**Loss Functions.** To help with the residual learning process, we incorporate a prior guidance loss for the prediction of principal frequency components, the final loss is defined in Eq. 5. The forecast with prior guidance can be considered a multi-task optimization problem [12], where $\mathcal{L}_\phi^{nonstat}$ ensures $q_\phi$ accurately predict the non-stationary principal frequency component and $\mathcal{L}_{\theta,\phi}^{forecast}$ guarantees that both model optimizes along the overall forecast accuracy.

$$\phi, \theta = \arg\min_{\phi,\theta} \sum_t \left( \mathcal{L}_\phi^{nonstat}(\mathbf{Y}_t^{non}, \hat{\mathbf{Y}}_t^{non}) + \mathcal{L}_{\theta,\phi}^{forecast}(\mathbf{Y}_t, \hat{\mathbf{Y}}_t) \right) \tag{5}$$

Here, the mean square error loss is used for both loss functions.

## 4 Experiments

### 4.1 Experiment Setup

**Datasets.** We use eight popular datasets in multivariate time series forecasting as benchmarks, including: (1-4) **ETT** (Electricity Transformer Temperature) [3][47] records the oil temperature and load of the electricity transformers from July 2016 to July 2018. Four subsets are included in this dataset, where ETThs are sampled per hour and ETTms per 15 minutes. (5) **Electricity** [4] contains the electricity consumption of 321 clients from July 2016 to July 2019 per 15 minutes. (6) **ExchangeRate** [5] contains the daily exchange rates of 8 countries from 1990 to 2016. (7) **Traffic** [6] includes the hourly traffic load of San Francisco freeways recorded by 862 sensors from 2015 to 2016. (8) **Weather** [7] is made up of 21 indicators of weather, including air temperature and humidity collected every 10 minutes in 2021.

For preprocessing, we apply z-score normalization [12] on all datasets to scale different variables to the same scale. Note that z-score normalization is unable to handle non-stationary time series since the statistics remain unchanged for different input instances [17]. The split ratio for training, validation, and test sets is set to 7:2:1 for all the datasets. We report datasets properties in Table 1, including (1) Trend Variation: Differences in the means across different sections of the dataset. (2) Seasonality Variation: We report the average variance over the Fourier spectrum to examine the presence of evolving seasonality. Other dataset details can be found at Appendix B.

Table 1: Properties of datasets and used hyperparameter $K$ of each dataset.

| Datasets | ETTh1 | ETTh2 | ETTm1 | ETTm2 | ExchangeRate | Electricity | Traffic | Weather |
|---|---|---|---|---|---|---|---|---|
| Trend Variation | 3.839 | 0.154 | 0.030 | 0.196 | 0.249 | 0.242 | 0.068 | 0.028 |
| Seasonality Variation | 3.690 | 1.013 | 3.330 | 1.648 | 0.435 | 2.645 | 14.225 | 0.387 |
| K | 4 | 3 | 11 | 5 | 2 | 3 | 30 | 2 |

**Evaluation.** We set the prediction length $H \in \{96, 168, 336, 720\}$, covering both short- and long-term rediction [30]. A fixed input window length $L = 96$ is used for all datasets. We evaluate the performance of baselines using mean squared error (MSE) and mean absolute error (MAE). The MSE and MAE are computed on z-score normalized data to measure different variables on the same scale. We report the final results on the test set for the model that performed optimally on the validation set.

**Backbone Models.** FAN is model-agnostic and could be applied to any prediction backbones. To validate its effectiveness, four state-of-the-art time-series forecasting model are used: MLP-based DLinear [46], Transformer-based Informer [47] and FEDformer[48], and convolutional neural network-based SCINet [25]. Notably, FEDformer also employs the Fourier transform for analyzing

---

[3]`https://github.com/zhouhaoyi/ETDataset`

[4]`https://archive.ics.uci.edu/ml/datasets/ElectricityLoadDiagrams20112014`

[5]`https://github.com/laiguokun/multivariate-time-series-data`

[6]`http://pems.dot.ca.gov`

[7]`https://www.bgc-jena.mpg.de/wetter/`

Table 2: Forecasting errors with and without FAN. The bold values indicate the best performance.

| Methods Metrics | DLinear | | +FAN | | FEDformer | | +FAN | | Informer | | +FAN | | SCINet | | +FAN | |
|---|---|---|---|---|---|---|---|---|---|---|---|---|---|---|---|---|
| | MAE | MSE | MAE | MSE | MAE | MSE | MAE | MSE | MAE | MSE | MAE | MSE | MAE | MSE | MAE | MSE |
| **ETTm2** 96 | 0.203 | 0.080 | **0.198** | **0.078** | 0.208 | 0.082 | **0.194** | **0.074** | 0.226 | 0.091 | **0.198** | **0.077** | 0.206 | 0.079 | **0.198** | **0.078** |
| 168 | 0.220 | 0.093 | **0.219** | **0.093** | 0.249 | 0.116 | **0.220** | **0.093** | 0.251 | 0.112 | **0.219** | **0.092** | 0.226 | 0.094 | **0.218** | **0.093** |
| 336 | 0.245 | 0.114 | **0.241** | **0.113** | 0.282 | 0.143 | **0.272** | **0.131** | 0.283 | 0.140 | **0.245** | **0.114** | 0.262 | 0.122 | **0.241** | **0.113** |
| 720 | 0.270 | 0.142 | **0.264** | **0.139** | 0.308 | 0.174 | **0.275** | **0.145** | 0.347 | 0.212 | **0.287** | **0.154** | 0.297 | 0.153 | **0.264** | **0.139** |
| **Electricity** 96 | 0.277 | 0.195 | **0.269** | **0.184** | 0.298 | 0.183 | **0.243** | **0.148** | 0.376 | 0.277 | **0.250** | **0.153** | 0.296 | 0.188 | **0.261** | **0.168** |
| 168 | 0.272 | 0.183 | **0.268** | **0.178** | 0.305 | 0.191 | **0.251** | **0.154** | 0.371 | 0.269 | **0.257** | **0.156** | 0.306 | 0.196 | **0.258** | **0.163** |
| 336 | 0.294 | 0.197 | **0.289** | **0.192** | 0.312 | 0.194 | **0.272** | **0.167** | 0.377 | 0.273 | **0.273** | **0.167** | 0.330 | 0.214 | **0.278** | **0.175** |
| 720 | 0.333 | 0.233 | **0.325** | **0.227** | 0.330 | 0.213 | **0.300** | **0.189** | 0.401 | 0.311 | **0.306** | **0.194** | 0.352 | 0.240 | **0.312** | **0.204** |
| **Exchange** 96 | **0.164** | **0.052** | 0.167 | 0.053 | 0.260 | 0.112 | **0.186** | **0.062** | 0.532 | 0.412 | **0.189** | **0.066** | 0.218 | 0.085 | **0.169** | **0.055** |
| 168 | 0.219 | 0.090 | **0.217** | **0.088** | 0.312 | 0.163 | **0.222** | **0.090** | 0.582 | 0.491 | **0.257** | **0.128** | 0.266 | 0.126 | **0.221** | **0.093** |
| 336 | **0.288** | **0.155** | 0.297 | 0.162 | 0.456 | 0.338 | **0.336** | **0.198** | 0.721 | 0.847 | **0.333** | **0.191** | 0.337 | 0.203 | **0.303** | **0.167** |
| 720 | 0.453 | 0.352 | **0.406** | **0.292** | 0.669 | 0.661 | **0.436** | **0.329** | 0.889 | 1.210 | **0.513** | **0.474** | 0.502 | 0.430 | **0.439** | **0.345** |
| **Traffic** 96 | 0.387 | 0.504 | **0.334** | **0.403** | 0.348 | 0.383 | **0.326** | **0.371** | 0.350 | 0.428 | **0.314** | **0.364** | 0.399 | 0.471 | **0.344** | **0.393** |
| 168 | 0.588 | 0.804 | **0.334** | **0.414** | 0.366 | 0.422 | **0.336** | **0.391** | 0.366 | 0.457 | **0.324** | **0.383** | 0.377 | 0.443 | **0.348** | **0.403** |
| 336 | 0.380 | 0.504 | **0.346** | **0.437** | 0.383 | 0.452 | **0.348** | **0.414** | 0.414 | 0.555 | **0.356** | **0.427** | 0.384 | 0.459 | **0.360** | **0.426** |
| 720 | 0.407 | 0.532 | **0.372** | **0.472** | 0.391 | 0.465 | **0.372** | **0.454** | 0.656 | 1.002 | **0.397** | **0.482** | 0.401 | 0.490 | **0.377** | **0.454** |
| **Weather** 96 | 0.249 | 0.180 | **0.214** | **0.173** | 0.368 | 0.299 | **0.252** | **0.187** | 0.299 | 0.221 | **0.221** | **0.175** | 0.265 | 0.199 | **0.215** | **0.170** |
| 168 | 0.284 | 0.237 | **0.254** | **0.210** | 0.409 | 0.358 | **0.304** | **0.240** | 0.363 | 0.320 | **0.258** | **0.215** | 0.305 | 0.245 | **0.256** | **0.208** |
| 336 | 0.344 | 0.304 | **0.298** | **0.275** | 0.463 | 0.459 | **0.366** | **0.321** | 0.439 | 0.437 | **0.323** | **0.297** | 0.341 | 0.310 | **0.304** | **0.270** |
| 720 | 0.380 | 0.358 | **0.345** | **0.340** | 0.495 | 0.526 | **0.441** | **0.432** | 0.496 | 0.524 | **0.368** | **0.360** | 0.383 | 0.371 | **0.340** | **0.322** |

seasonality. Results later show that FAN continues to make considerable improvements over these frequency-based solutions like FEDformer.

**Implementation and Settings.** For the non-stationary prediction module $q_\phi$ in FAN, the MLP model has hidden sizes [64, 128, 128]. All the experiments are implemented by PyTorch [34] and are conducted for five runs with fixed seeds $\{1, 2, 3, 4, 5\}$ on NVIDIA RTX 4090 GPU (24GB). For the different baselines, we follow the implementation and settings provided in their official code repository. ADAM [18] as the default optimizer across all the experiments. More experiment details, including training details and hyperparameter, can be found in Appendix A.1.

**Selections of Hyperparameter $K$.** FAN allows for $K$ to be any integer number less than $L$. Regarding the selection of $K$, we analyze these benchmarks and found that the main variation frequencies of these datasets are within 10% of the maximum amplitude. Therefore, we select the value of $K$ based on the average maximum amplitude within 10% in the training set, the selected $K$ is shown in Table 1. More evidence of this selection strategy is at Sec. 4.4, and we provide a detailed hyperparameter sensitivity analysis at Appendix D.1.

## 4.2 Main Results

We report MAE/MSE forecasting errors of the baselines and FAN in Table 2. Since the performance in ETT datasets shows similar results, only results of ETTm2 are reported. The full results of the ETT benchmarks and further discussion are in Appendix E.2.

As shown in Table 2, our proposed FAN effectively enhances the performance of all four backbone models, by a large margin, achieving state-of-the-art performance on five datasets. Specifically, on the ETTm2, Electricity, Exchange, Traffic, and Weather datasets, the average MSE performance improvements are rather significant: 10.81%, 21.49%, 51.27%, 21.97%, and 21.55% respectively. It clearly shows that frequency residual learning of FAN effectively mitigates the impacts of evolving seasonal and trend patterns and enhances the stationarity that simplifies the prediction for backbones.

FAN demonstrates increasing performance improvements as the prediction length extends in the Informer backbone, with MSE improvements of 9.87%, 18.87%, 36.91%, 16.26%, and 20.05%, from 96 steps to 720 steps. We believe this can be attributed to the fact that the periodicity contained in the prediction series increases with step length, and the FAN's pattern prediction module helps uncover periodicity in longer step lengths, thereby enhancing long-term prediction effectiveness. It is important to note that even in the models that utilize FFT to analyze seasonal patterns, like FEDformer, we still observe significant performance improvements (19.81%). This conclusion underscores our model's advantage in handling non-stationary aspects by directly extracting non-stationary seasonality patterns rather than learning these patterns.

## 4.3 Comparison With Reversible Instance Normalization Methods

In this section, we compare FAN with three state-of-the-art normalization methods for non-stationary time series forecasting: SAN [28], Dish-TS [10], and RevIN [17], with the same experimental setup as Sect. 4.2. We report the average MSE over all the forecasting lengths of all backbones for all datasets in Table 3. It is evident that FAN generally outperforms the baseline models, except for a few cases with a close margin. Here, SAN generally ranks second as it slices the whole sequence into sub-series which can make seasonal patterns into evolving trends that could be partially predicted with its statistics prediction module. In comparison, RevIN and Dish-TS have much worse performance. Detailed results of all cases and further discussions are provided in Appendix E.4.

Table 3: The MSE performance averaged across all steps. Bold values indicate the best performance.

| Models | DLinear | | | | FEDformer | | | | Informer | | | | SCINet | | | |
|---|---|---|---|---|---|---|---|---|---|---|---|---|---|---|---|---|
| Methods | FAN | SAN | Dish-TS | RevIN | FAN | SAN | Dish-TS | RevIN | FAN | SAN | Dish-TS | RevIN | FAN | SAN | Dish-TS | RevIN |
| ETTh1 | **0.441** | 0.454 | 0.465 | 0.477 | **0.443** | 0.530 | 0.565 | 0.591 | **0.465** | 0.624 | 0.714 | 0.688 | **0.442** | 0.454 | 0.489 | 0.472 |
| ETTh2 | 0.135 | **0.134** | 0.136 | 0.149 | 0.149 | **0.148** | 0.217 | 0.183 | **0.164** | 0.201 | 0.259 | 0.199 | **0.136** | 0.139 | 0.160 | 0.149 |
| ETTm1 | 0.395 | **0.390** | 0.405 | 0.419 | **0.400** | 0.416 | 0.489 | 0.491 | **0.397** | 0.427 | 0.504 | 0.485 | 0.395 | **0.393** | 0.424 | 0.443 |
| ETTm2 | **0.105** | 0.106 | 0.108 | 0.113 | 0.111 | **0.106** | 0.125 | 0.121 | **0.106** | 0.114 | 0.153 | 0.130 | **0.105** | **0.105** | 0.122 | 0.112 |
| Electricity | **0.193** | 0.200 | 0.201 | 0.207 | **0.164** | 0.169 | 0.181 | 0.180 | **0.167** | 0.191 | 0.219 | 0.190 | 0.177 | 0.175 | 0.207 | **0.164** |
| Exchange | **0.149** | 0.172 | 0.265 | 0.190 | **0.170** | 0.192 | 0.333 | 0.267 | **0.168** | 0.265 | 0.472 | 0.238 | 0.162 | 0.174 | 0.281 | 0.183 |
| Traffic | **0.432** | 0.514 | 0.591 | 0.652 | 0.408 | **0.395** | 0.433 | 0.424 | **0.400** | 0.515 | 0.446 | 0.894 | 0.419 | 0.431 | 0.489 | 0.442 |
| Weather | **0.249** | 0.250 | 0.269 | 0.272 | 0.295 | **0.272** | 0.562 | 0.280 | **0.254** | 0.256 | 0.322 | 0.275 | 0.242 | **0.242** | 0.250 | 0.251 |

**Show Cases.** Fig. 3 illustrates the forecasting results with DLinear backbone in Traffic to show why FAN has performance advantages. This data has very clear evolving seasonality with daily waveform patterns. FAN can extract trends and seasonal patterns especially the seasonal patterns during weekends while Dish-TS and RevIN only focus on trends statistics. Furthermore, FAN can adaptively adjust frequency pattern forecasting results based on the input main frequency signals, capturing the evolving patterns between the input and horizon. Fig. 3(a) clears shows FAN can identify the seasonal patterns with increasing amplitudes from hour 100~150.

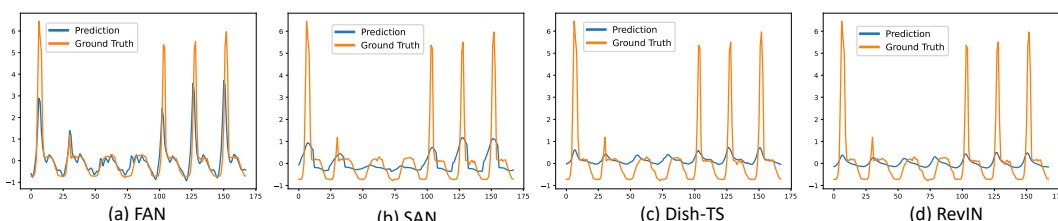

| (a) FAN | (b) SAN | (c) Dish-TS | (d) RevIN |

Figure 3: Visualization of long-term 168 steps forecasting results of a test sample in Traffic dataset, using DLinear enhanced with different normalization methods.

**Stationarity Analysis.** To verify our model's effectiveness against non-stationarity, we use the ADF test [39] to examine the stationarity of the data after normalization. The results are shown in Fig. 4(a), smaller value (further from the center) indicates higher stationarity. Compared to previous normalization methods, our model achieves greater stationarity across all datasets, particularly in cases with larger seasonal patterns (Traffic, ETTh1, ETTm1). In some datasets, e.g., Weather, Exchange, despite having less apparent seasonality, our model still enhances stationarity. We attribute this to our method's ability to adaptively capture the low-frequency trend signals, such as mixed-linear changes, while other methods assume consistent distribution over a period and remove estimated statistics, due to which they might fail to capture these intricate trend patterns.

**Model Efficiency.** We compare the performance, training time per iteration, and number of parameters with SAN on Traffic with $D = 862, H = 720$. DLinear is used for both as the backbone. The results are shown in Fig. 4(b). FAN and SAN have similar training iteration times, but FAN has $29.79\%$ less parameters. Moreover, FAN achieves a $15.56\%$ improvement in MSE and a $15.30\%$ improvement in MAE. This highlights our model's effectiveness and efficiency.

**Various Input Length.** In time series learning, the non-stationarity of inputs varies with the choice and change of the time window [35], which in turn impacts the performance of deep learning models [27]. Therefore, we compare the performance changes under different input lengths on

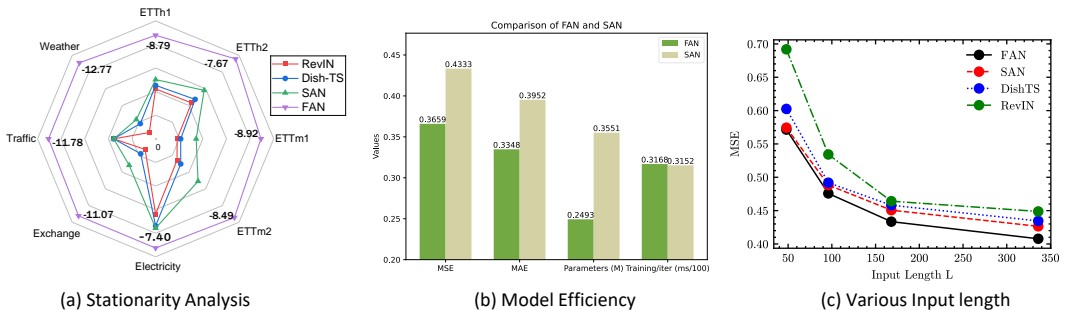

| (a) Stationarity Analysis | (b) Model Efficiency | (c) Various Input length |

Figure 4: Comparison with other normalization methods. (a) ADF test after normalization, the smaller the value, the higher the stationarity. (b) Model efficiency comparison with SAN, including MSE/MAE, parameters (in millions), and training time per iteration (ms/100). (c) Performance in MSE vs. input length on the ETTm2 dataset.

ETTm2 dataset and DLinear as the backbone. Fig. 4(c) shows FAN exhibits the best performance across all windowed data. As can be seen, compared to other models, as the input length increases, among these normalizations, the enhancement of increases the most. The MSE enhancement with previous SOTAs increases from 0.49% in short inputs $L = 48$ to 4.37% in long inputs with $L = 336$. This demonstrates that the instance-wise DFT is capable of extracting more seasonal patterns from the longer input windows.

## 4.4 TopK vs. Frequency Distributions

As different datasets might have different non-stationary patterns, it is crucial to select appropriate K frequency components from inputs. We study the relations between the selected TopK and the frequency distribution on the ExchangeRate and Traffic datasets.

Fig. 5 plots the frequency amplitude distribution for frequency 0~32 by performing DFT towards the different input instances with $L = 96$ of the whole training sequences. Here, we can see the clear relation between the selection of K and the frequency amplitude distributions. As we can see, the Traffic dataset contains rich seasonal signals ranging from 0~32 while the ExchangeRate dataset only has mainly one principal frequency component with frequency 0 (trend) in the inputs. Thus, the prediction on the ExchangeRate dataset might not benefit from a bigger K while a bigger K indeed helps for the Traffic dataset. Results for more datasets are in Appendix B.

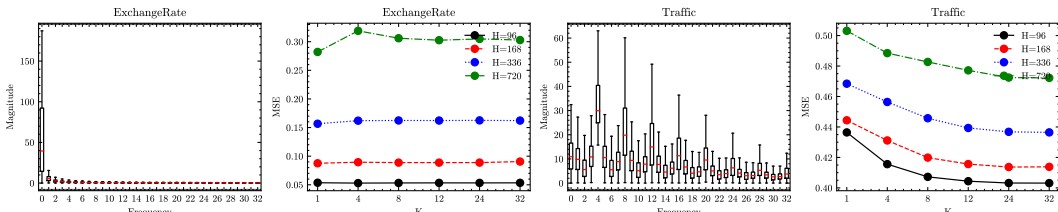

Figure 5: Frequency distributions vs. forecast error in MSE with different K and output length $H$.

## 4.5 Ablation Studies

**Main Components.** This section aims to evaluate the effectiveness of various FAN's designs. Three variants are studied: "w/o predict" denotes the removal of the non-stationary pattern prediction module and directly reconstructing $\hat{\mathbf{Y}}^{non}$ with $\mathbf{X}^{non}$. "pure backbone" refers to the omission of the reconstruction in the output or "w/o backbone" is the omission of the stationary part. We evaluate their performance on two non-stationary datasets, ETTh1 and Weather. The experimental settings are consistent with those described in Section 4.2. The evaluation results, along with the standard deviations, are presented in Table 4. The results show that FAN achieves best performance across all metrics in all variants. *FAN w/o backbone* ranks second as the learning model of FAN already learns the principle changes. In comparison, the results from *pure backbone* is the weakest, as it cannot handle nonstationary signals. *FAN w/o predict* also has poor performance. Those results clearly

Table 4: Forecasting errors under the multivariant setting with respect to variations of FAN with SCINet backbone. The best performances are highlighted in bold.

| Variations Datasets | Steps Metrics | FAN | | w/o predict | | pure backbone | | w/o backbone | |
|---|---|---|---|---|---|---|---|---|---|
| | | MAE | MSE | MAE | MSE | MAE | MSE | MAE | MSE |
| ETTh1 | 96 | **0.427±0.000** | **0.362±0.001** | 0.517±0.001 | 0.582±0.001 | 0.527±0.022 | 0.549±0.071 | 0.457±0.001 | 0.392±0.001 |
| | 168 | **0.454±0.003** | **0.395±0.002** | 0.536±0.000 | 0.610±0.001 | 0.614±0.031 | 0.603±0.097 | 0.494±0.002 | 0.436±0.003 |
| | 336 | **0.487±0.004** | **0.439±0.003** | 0.557±0.001 | 0.654±0.001 | 0.621±0.022 | 0.622±0.079 | 0.524±0.002 | 0.474±0.004 |
| | 720 | **0.571±0.004** | **0.572±0.003** | 0.632±0.003 | 0.789±0.007 | 0.633±0.013 | 0.636±0.065 | 0.616±0.009 | 0.618±0.013 |
| Weather | 96 | **0.215±0.002** | **0.170±0.001** | 0.293±0.002 | 0.271±0.001 | 0.335±0.007 | 0.332±0.008 | 0.246±0.000 | 0.195±0.000 |
| | 168 | **0.253±0.001** | **0.206±0.001** | 0.325±0.003 | 0.310±0.005 | 0.347±0.007 | 0.346±0.013 | 0.284±0.001 | 0.232±0.001 |
| | 336 | **0.299±0.001** | **0.268±0.002** | 0.368±0.002 | 0.376±0.003 | 0.376±0.010 | 0.370±0.005 | 0.329±0.001 | 0.296±0.001 |
| | 720 | **0.339±0.003** | **0.322±0.002** | 0.411±0.004 | 0.441±0.004 | 0.411±0.008 | 0.401±0.010 | 0.367±0.002 | 0.341±0.003 |

show that trends and seasonal patterns do evolve and that our proposed residual frequency learning is crucial in dealing with these changes.

**Instance-wise vs. Global Fourier Analysis.** This section investigates the effect from instance-wise Fourier Analysis of FAN. Previous Fourier-based methods select predominant Fourier signals based on fixed frequencies [41, 27, 43, 9]. However, as shown in Fig. 6, on the Traffic and Electricity datasets, the predominant components from the input-wise view are not fixed but exhibit distinct distribution characteristics and vary across the inputs. The assumption of fixed spectrum and the reality of changing frequency limits their performance, supported with two additional experiments on Fourier-based backbones at Appendix E.3.

Here, we compare the performance of FAN with fixed frequencies computed using global sequences and original FAN with instance-specific frequencies. The results are shown in Table 5.

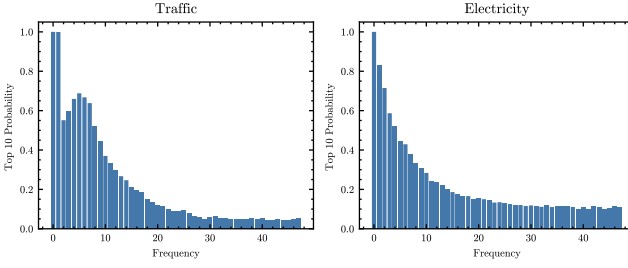

Figure 6: Top 10 selection propablity density on Traffic and Electricity datasets.

Table 5: MSE Performance between instance-wise (FAN) and global selection (Fixed) on SCINet backbone.

| | Electricity | | | | |
|---|---|---|---|---|---|
| Steps | 96 | 168 | 336 | 720 | Avg.Imp. |
| FAN | **0.162** | **0.165** | **0.173** | **0.194** | 18.50% |
| Fixed | 0.176 | 0.192 | 0.231 | 0.265 | - |
| | Traffic | | | | |
| Steps | 96 | 168 | 336 | 720 | Avg.Imp. |
| FAN | **0.393** | **0.403** | **0.426** | **0.454** | 10.29% |
| Fixed | 0.446 | 0.457 | 0.469 | 0.496 | - |

As shown in Table 5, by selecting instance-wise predominant frequencies, FAN achieves an average improvement of 18.50% and 10.29% on the Electricity and Traffic datasets respectively. This highlights instance-wise frequency selection rather than assuming fixed frequency patterns.

# 5 Conclusion

In this paper, we study the problem of non-stationary time series prediction. We identify the fact that traditional statistical measurement-based instance-wise normalization can not effectively recover the evolving seasonal patterns. We propose FAN to perform instance normalization for each input window. The Fourier transform is used to remove the main frequency components in the inputs and reconstruct the Fourier basis through denormalization. To address the evolving trend and seasonal patterns between inputs and outputs, we utilize a simple MLP model to predict the changes in the extracted non-stationary pattern. The effectiveness of FAN is verified with a set of experiments on eight widely used benchmark datasets. Compared to other state-of-the-art normalization baselines, FAN significantly improves the prediction performance and outperforms state-of-the-art normalization methods. One potential avenue for improvement involves the autonomous determination of an optimal K for the selection of principal frequency components.

## Acknowledgments and Disclosure of Funding

This work is supported in part by the National Natural Science Foundation of China (Grant No. 6247075381). We would also like to thank the anonymous reviewers for their constructive feedback and suggestions.

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

# A  Reproducibility

## A.1  Experiment Details

We make our code publicly available 2, including the backbones and baselines; the backbones and baselines code are based on their official public GitHub repositories and we use the default parameter settings or the optimal parameter settings in their paper. We used a batch size of 32, a learning rate of 0.0003, and trained each run for 100 epochs, with an early stopper set to patience as 5. For the experimental results, $K$ is set as the number of frequencies greater than 10% of the maximum amplitude. For a fair comparison, other normalization methods were also tuned accordingly to ensure optimal results, only the normalization hyperparameters were tuned, and no other experiment parameters are tuned during the experiment phase.

## A.2  Pseudocode of GPU-Friendly Normalization

```python
def norm(x, k):
    # x: (BxNxL) multivariate time series batch input
    # k: hyper parameter selecting k largest magnitude frequencies

    # applying fourier transform to each series O(Llog(L))
    z = torch.fft.rfft(x, dim=2)

    # find top k indices O(L + k)
    ks = torch.topk(z.abs(), k, dim = 2)
    top_k_indices = ks.indices

    # top-k-pass filter O(L + k)
    mask = torch.zeros_like(z)
    mask.scatter_(2, top_k_indices, 1)
    z_m = z * mask

    # applying inverse fourier transform to each fourier series O(Llog(L))
    x_m = torch.fft.irfft(z_m, dim=2).real
    x_n = x - x_m
    return x_n
```

Listing 1: GPU-Friendly Implimentation of FRL

# B  Dataset Details

## B.1  Fourier Amplitude Distribution

Frequency amplitude variation and composition are closely related to the non-stationary pattern shift [20]. To analyse its effect, we use $L = 96$ to plot the frequency amplitude distribution of all input windows used in our eight benchmarks in Fig. 7.

In Fig. 7, it can be clearly observed that in many datasets such as ETTh1, ETTh2, ETTm1, Traffic, and Electricity datasets, besides the low-frequency trend patterns, the high-frequency parts also exhibit significant variations, especially in ETTh1, ETTm1, and Traffic datasets. This may also be the reason for our significant improvements on these datasets (with maximum improvements of 19.90%, 7.02%, and 18.65% respectively). However, in the ETTh2, ExchangeRate, and Weather datasets, which have relatively low seasonal variation, our model's improvement compared to state-of-the-art methods is relatively smaller. This is naturally because these datasets do not contain much seasonal non-stationary information for further improvement.

## B.2  Main Frequency Density

FAN select top $K$ amplitude signals, compared to previous methods based on Fourier transform, we do not use a fixed frequency set [41, 27] or randomly select [48] the frequencies. This is aligned with our observations in real data: the principal frequency signals may have a distinct distribution,

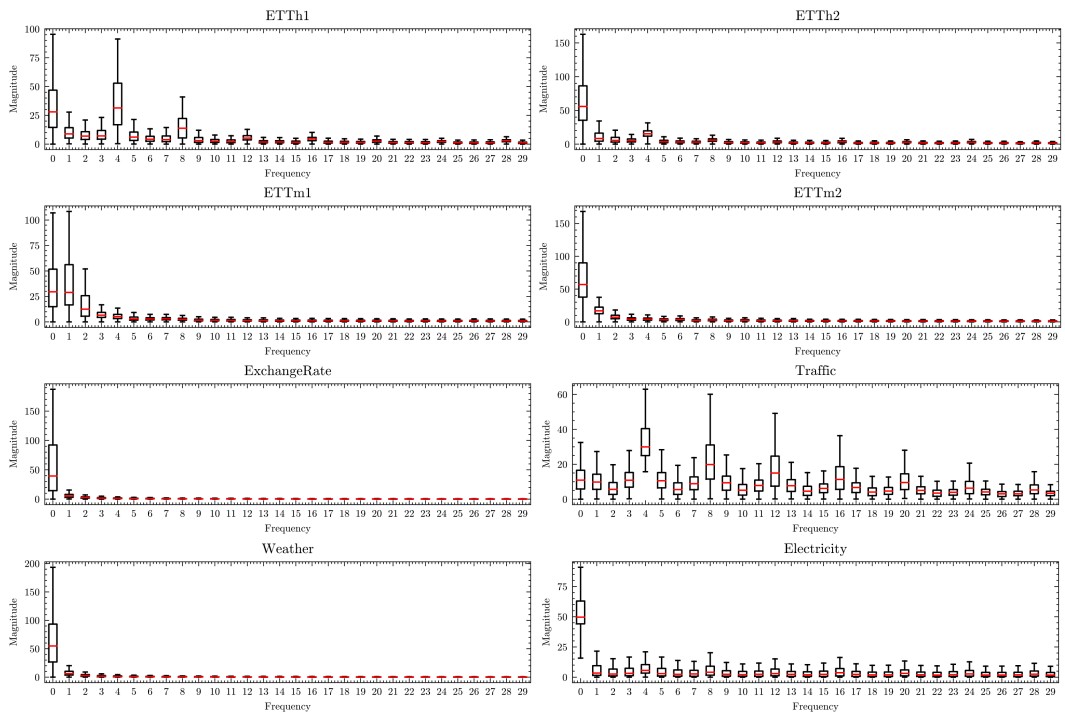

Figure 7: The distribution of the various Fourier components of the data, we display the first 30 frequencies.

rather than being composed solely of fixed or pure random frequency signals. We plot the probability of input frequencies being selected into the top 10 signals in the input signal, as shown in Fig. 8. Although the low-frequency trend signals dominate in amplitude, many high-frequency signals still play a dominant role in some inputs, highlighting the importance of considering the entire spectrum, not just the low/high or random selected frequencies. Furthermore, this analysis shows that there may be significant differences of the main frequency components between different inputs.

However, previous methods based on the Fourier transform assumed that the main frequency signal is constant across inputs [41, 27]. In contrast, our model can dynamically extract Fourier-based signals from the inputs which enables better extraction of seasonal information, especially when the input patterns vary a lot.

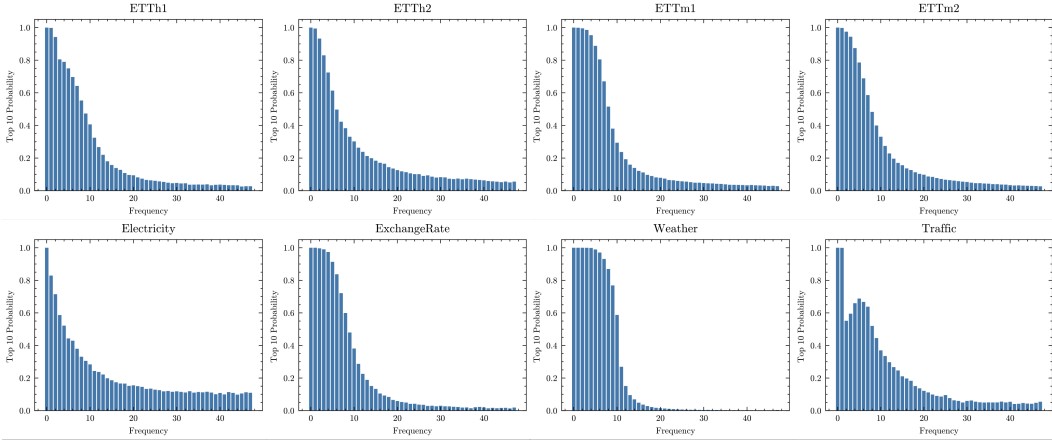

Figure 8: Probability density of frequencies get selected in the top 10 removal process, we use an input length $L = 96$ as the analysis length.

## B.3 Variation of Main/Residual Components

We examine the relative variations of the normalized main and residual components in Fig. 9. The quantitative results are obtained by calculating the relative amplitude variations of the Fourier components between the input and output in the frequency domain. In particular, across all benchmarks, the variations in the main frequency components are smaller than those in the residuals. We believe that this is the reason why a simple MLP is effective enough to capture the main frequency variations, as its shift is relatively small, as shown in the Appendix D.2.

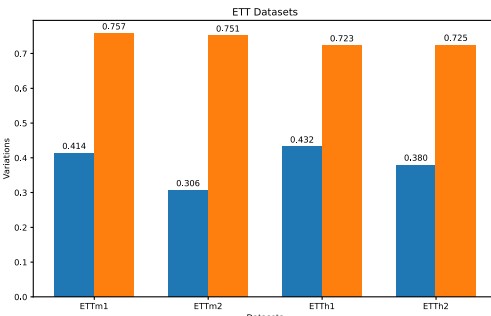 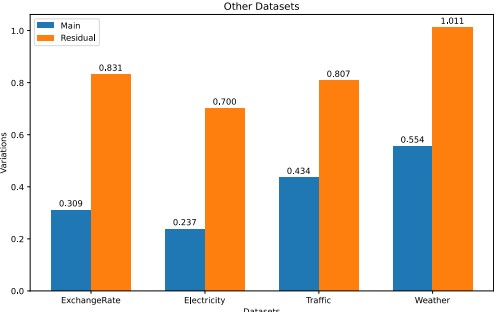

Figure 9: The relative changes in amplitude of the main/residual frequency components in the time domain. The results are evaluated using input length $L = 96$ and averaged across the whole dataset.

## B.4 Trend/Seasonal Variation

We explain more details of how the trend and seasonal variation in Table 1 are calculated.

**Trend Variation** To capture global trend shifts, we calculate the mean values over different regions of the dataset. Specifically, given a timeseries dataset $\mathcal{X} \in \mathbb{R}^{N \times D}$, we first chronologically split it into $\mathcal{X}^{\text{train}}$, $\mathcal{X}^{\text{val}}$, and $\mathcal{X}^{\text{test}}$, representing the training, validation, and testing datasets, respectively. The trend variations are then computed as follows:

$$\text{Trend Variation} = \left| \frac{\text{Mean}_N(\mathcal{X}^{\text{train}}) - \text{Mean}_N(\mathcal{X}^{\text{val,test}})}{\text{Mean}_N(\mathcal{X}^{\text{train}})} \right| \qquad (6)$$

where the subscripts indicate the dimension of mean, $|\cdot|$ denotes the absolute value operation, and $\mathcal{X}^{\text{val,test}}$ represents the concatenation of the validation and test sets. Note that, to obtain relative results across different datasets, the trend variation is normalized by dividing by the mean of the training dataset. We obtain the first dimension to be the value inthe main content Table 1.

**Seasonal variations.** We evaluate seasonal changes by analyzing the variations in Fourier frequencies across all input instances. Given the inputs, $X \in \mathbb{R}^{N_i \times L \times D}$ where $N_i$ is the number of inputs. We first obtain the FFT results of all inputs, denoted as $Z \in \mathbb{C}^{N_i \times L \times D}$. Then, we calculate the variance across different inputs and normalize this variance by dividing by the mean of each input as:

$$\text{Seasonal Variation} = \frac{\text{Var}_{N_i}[\text{Amp}(Z)]}{\text{Mean}_L(X)} \qquad (7)$$

where the subscripts indicate the dimension of the operation. We sum the results across all channels for the value in Table 1.

## C Theoritical Analysis

This section discusses the effect of FAN on stationarity and temporal distribution in a theoretical perspective. We conclude that FAN enhances the stationarity of the input and mitigates distribution in the time domain.

### C.1 Preliminary

**Discrete Fourier Transform.** Given a multivariate time series input $\mathbf{X}$, we independently apply the 1-dim Fourier transform to each dimension $\mathbf{x}^{(i)}$, hence, we illustrate in vector settings. For a discrete

time series vector $\mathbf{x} \in \mathbb{R}^L$ with $L$ time steps, it is transformed into Fourier domain by applying the 1-dim DFT, and can be transformed back using 1-dim IDFT, which can be defined as:

$$\text{DFT}: \quad \mathbf{z}[w] = \sum_{t=0}^{L-1} \mathbf{x}[t] \cdot e^{-i2\pi \frac{wt}{L}}$$

$$\text{IDFT}: \quad \mathbf{x}[t] = \frac{1}{L} \sum_{w=0}^{T-1} \mathbf{z}[w] \cdot e^{i2\pi \frac{wt}{L}}$$

(8)

where $w$ is current frequency, $t$ is current time step, and $\mathbf{z}$ represents the Fourier transformation results which is a complex vector with real and imaginary parts, the amplitude and phase can be calculated as:

$$\text{Mag}: \mathbf{a}[w] = \frac{\sqrt{\text{Re}(\mathbf{z}[w])^2 + \text{Im}(\mathbf{z}[w])^2}}{L}$$

$$\text{Pha}: \mathbf{p}[w] = \operatorname{atan} 2(\text{Im}(\mathbf{z}[w]), \text{Re}(\mathbf{z}[w]))$$

(9)

where $\text{Im}(\mathbf{z}[m])$ and $\text{Re}(\mathbf{z}[m])$ indicate imaginary and real parts of a complex number, and atan2 is the two-argument form of arctan.

**Distribution Of Fourier Components.** The distributions of Fourier amplitude and phase can be modeled as Rayleigh distribution and uniform distribution respectively [14], thus the probabilistic density function can be represented as:

$$f(a, p) = \text{Rayleigh}\,(a \mid \sigma) \cdot \text{U}(p \mid 0, 2\pi)$$

$$= \frac{a}{2\pi\sigma^2} \cdot \exp\left(-\frac{a^2}{2\sigma^2}\right)$$

$$(a \geq 0, 0 \leq p \leq 2\pi).$$

(10)

where a and p denotes a amplitude and phase scalar variable, $\sigma$ is the scale parameter of the distribution. Thus, the frequency domain distribution ($T$ non-identically-distributed variables) can be modeled as a joint Rayleigh distribution with different scale parameters:

$$f(\mathbf{a}, \mathbf{p}) = \text{Rayleigh}\,(\mathbf{a} \mid \boldsymbol{\sigma}) \cdot \text{U}(\mathbf{p} \mid 0, 2\pi)$$

(11)

## C.2 Variance Over Spectrum

Along with the time series spectral theory [35], a time series with smaller variance in the spectrum is more stationary, in this section, we try to prove the proposed FAN can reduce the variance over spectrum, thus enhance the stationarity of the input data. Hence, we prove that, given an univariate time series real value vector $\mathbf{x} \in \mathbb{R}^T$, after removing main frequency components $\mathbf{z}[k] \in \mathcal{K}$, the variance on spectrum can be reduced $\text{Var}\,(\mathbf{a}^{res}) < \text{Var}\,(\mathbf{a})$.

Here, the marginal distribution of the amplitude vector (the spectrum) $\mathbf{a}$ is represented as a joint Rayleigh distribution with different scale parameters:

$$f(\mathbf{a}) = \int f(\mathbf{a}, \mathbf{p}) d\mathbf{p}$$

$$= \prod_{i=1}^{L} \frac{a}{\sigma_i^2} \cdot \exp\left(-\frac{a^2}{2\sigma_i^2}\right)$$

(12)

Note that although we assume that the frequency components are independent with each other, this assumption is actually widely used [16] since it is quite possible that a specific component changes independently, e.g., the daily weekly changes while the monthly periodicity stays the same. Following the principle of additivity of variance for independent variables [13], the variance of the amplitude vector $\mathbf{a}$ can be expressed as follows:

$$\text{Var}\,(\mathbf{a}) = \sum_{i}^{L} \frac{4 - \pi}{2} \sigma_i^2$$

(13)

after removing frequencies $k \in \mathcal{K}$, the joint distribution actually becomes:

$$f(\mathbf{a}^{res}) = \prod_{i=1, i \notin \mathcal{K}}^{L} \frac{a}{\sigma_i^2} \cdot \exp\left(-\frac{a^2}{2\sigma_i^2}\right) \tag{14}$$

thus, the variance of the whole distribution after removing top $K$-amplitude signals reduces to a smaller number, since the independent variance of of each dimension is positive, which is:

$$\mathrm{Var}\left(\mathbf{a}^{res}\right) = \sum_{i=1, i \notin \mathcal{K}}^{L} \frac{4-\pi}{2}\sigma_i^2 < \mathrm{Var}\left(\mathbf{a}\right) \tag{15}$$

### C.3 Influence On Temporal Distribution

**Relation with Temporal Statistics.** The zero frequency of the Fourier transform divided by $L$ is actually the mean of the statistical measure, and the energy of the Fourier transform of frequency components above zero is equivalent to the variance of the input scaled by $L$, proved as follow:

$$\mathbb{E}[\mathbf{x}] = \frac{1}{L} \sum_{t=0}^{L-1} \mathbf{x}[t] = \frac{1}{L}\mathbf{z}[0] \tag{16}$$

According to Parseval theorem [5], for a discrete signal $\mathbf{x}$, its energy is identical in both the time and frequency domain:

$$\sum_{t=0}^{L-1} |\mathbf{x}[t]|^2 = \sum_{w=0}^{L-1} |\mathbf{z}[w]|^2 = L\mathbb{E}[\mathbf{x}^2] \tag{17}$$

Thus the variance of the input signal can be defined as the energy of Fourier components with frequency above zero.

$$\mathrm{Var}[\mathbf{x}] = \mathbb{E}[\mathbf{x}^2] - \mathbb{E}^2[\mathbf{x}]$$
$$= \frac{1}{L} \sum_{w=0}^{L-1} |\mathbf{z}[w]|^2 - \frac{1}{L^2}|\mathbf{z}[0]|^2 \tag{18}$$

$$\tag{19}$$

**Influence On Mean.** Since the mean is equal to the zero frequency component in time domain and for any other components, the expectation is zero since they are all number of periodic sin/cos signals, after removing the zero frequency component, the expectation is then equal to zero. Due to this property, this is also known as the 'detrending' in traditional signal processing [32], proved as:

$$\mathbb{E}[\mathbf{x}^{res}] = \mathbb{E}[\mathbf{x} - \mathrm{IDFT}(\mathbf{z}[0])] = \mathbb{E}[\mathbf{x} - \frac{1}{L}\mathbf{z}[0]] = \mathbb{E}[\mathbf{x} - \mathbb{E}[\mathbf{x}]] = 0 \tag{20}$$

**Influence On Variance.** Since the The normalization step select and remove top $K$ amplitude Fourier components, the Fourier spectrum energy will be significantly diminished, defined as:

$$\sum_{w=0, w \notin \mathcal{K}}^{L-1} |\mathbf{z}[w]|^2 \ll \sum_{w=0}^{L-1} |\mathbf{z}[w]|^2 \tag{21}$$

thus, the input variance then can be largely reduced, which is:

$$\mathrm{Var}[\mathbf{x}^{res}] \ll \mathrm{Var}[\mathbf{x}] \tag{22}$$

In summary, our method can effectively reduce the range of data distribution, which is crucial for enhancing the performance of the backbone model and minimizing the risk of overfitting [12].

## C.4 Fourier Spectrum Empirical Analysis

The variance in the Fourier spectrum is an important indicator reflecting stationarity [20]. The closer the frequency components are to each other, the smaller the variance between the components, thus the stronger the stationarity [35]. Therefore, we compare the changes in frequency domain components for different methods and present the results in Fig. 10. In Fig. 10, after FAN's normalization step, the distribution exhibits alignment of the input and output, and the range of the distribution mean has decreased to 8, compared with previous methods which are round 80, 70, 70 respectively for SAN, Dish-TS and RevIN. However, other methods still show significant differences between the input and output distributions, with the range of the frequency domain amplitude distribution reaching up to 80, indicating the presence of strong non-stationary signals. This highlights the effectiveness of our method in handling non-stationarity, especially for seasonal periodic signals, which previous methods have not successfully considered.

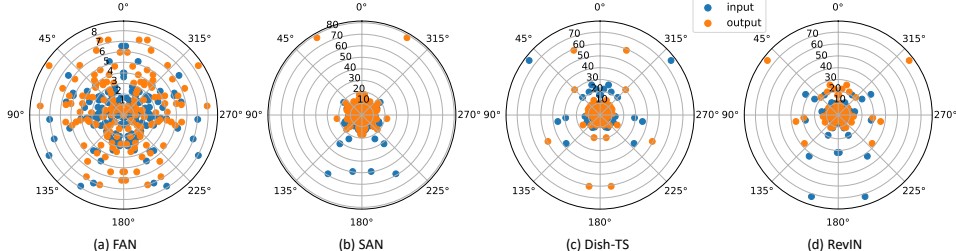

Figure 10: Fourier spectrum on polar axis of ETTm2 dataset with $L = 96$ after various normalization methods. Each point indicates one frequency component averaged across the dataset. The blue dots indicate the input Fourier components, the orange dots represent the output Fourier components. FAN remove top 5 Fourier components, and SAN slice in 12.

# D Ablation Study

## D.1 Hyperparameter Analysis

Our model incorporates a hyperparameter $K$, which represents the maximum frequency count selection. In this section, we provide a sensitivity analysis for this parameter in Fig. 11. We observe that our proposed FAN achieves stable performance across various parameter settings.

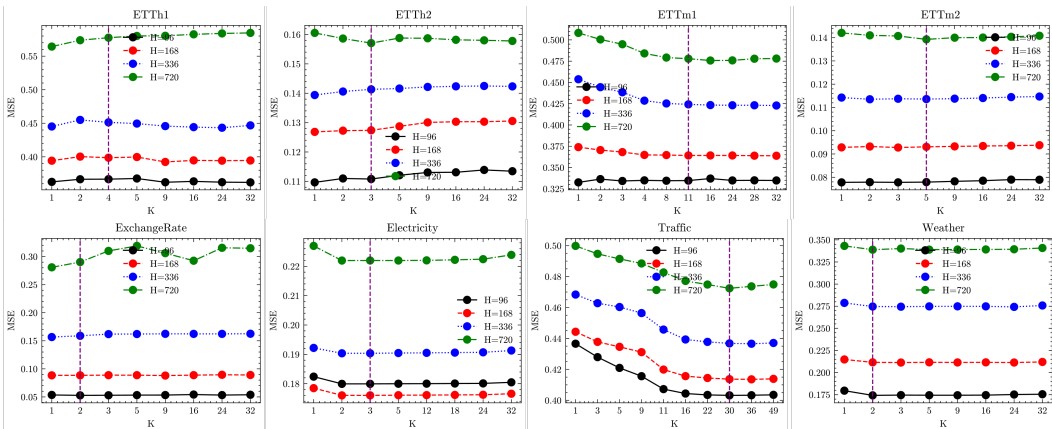

Figure 11: Sensitivity analysis of hyper-parameter $K$, and we select dataset-specific $K$ across the experiments. The purple line denote the selected $K$ through our 10% of largest magnitude selection rule. We use DLinear as backbone with and use MSE as the evaluation metric, other settings are identical with the main results settings.

Moreover, from Fig. 11, we note the following observations: (1) As the prediction length increases, the need for a larger $K$ becomes more apparent, significantly enhancing performance. This is likely

due to the increased prediction steps that bring more frequency information into the model. For example, in the ExchangeRate dataset, when the prediction length is 720, $K = 16$ outperforms using only 3 and 6 frequencies. (2) In datasets with higher sampling rates, such as Traffic and ETTm2, larger $K$ values enhance performance at all prediction steps. This could be attributed to the finer granularity of sampling in these datasets (minute-level) compared to yearly sampling in ExchangeRate and hourly in ETTh2, resulting in richer frequency signals in ETTm2 and Traffic, thereby enhancing FAN's performance on these datasets.

## D.2 Pattern Prediction Module

To justify our rather simple MLP structure for predicting the future main frequency component, we extended the basic MLP with three additional layers to observe the results. These layers include

- +MLP: adding an additional MLP layer on top of the basic MLP.
- +GRU [4]: adding a GRU layer, which is a recurrent neural network, mitigates the problem of gradient vanishing through the gating mechanism.
- +TSMixer [7]: adding a TSMixer layer, which is a state-of-the-art lightweight model that also considers inter-dimensional relationships.

We perform ablation on ETTh1, ExchangeRate, Weather datasets, under experiment settings of Section 4.2, we report the MAE/MSE evaluation metrics, and the result is shown at Table 6.

In Table 6, the three-layer MLP of FAN performed best overall on three datasets and more complex models tend to perform worse. We believe this is due to the following reasons: (1) The main frequency signal provides a baseline position for the backbone model, leading to more robust predictions and thus to greater model robustness. (2) The main frequency signal is subject to underlying physical characteristics, resulting in relatively slower changes. This has been verified by observations in Appendix B.3, showing that the main frequency signal changes at least 40.27% more slowly compared to the residual frequency signal. Therefore, a simple three-layer MLP is sufficient to provide effective and somewhat more robust predictions. However, a four-layer MLP and a GRU also achieved the best performance in some metrics, indicating that there is still room for improvement in future work.

Table 6: Multivariate Forecasting MAE/MSE results mean and standard deviation, the bold letter indicates the best performance.

| Methods | | FAN | | +MLP | | +GRU | | +TSMixer | |
|---|---|---|---|---|---|---|---|---|---|
| Dataset | Steps | MAE | MSE | MAE | MSE | MAE | MSE | MAE | MSE |
| ETTh1 | 96 | **0.427±0.000** | 0.362±0.001 | 0.427±0.000 | **0.362±0.001** | 0.482±0.005 | 0.424±0.008 | 0.463±0.009 | 0.403±0.012 |
| | 168 | **0.454±0.003** | **0.395±0.002** | 0.465±0.001 | 0.417±0.001 | 0.506±0.006 | 0.457±0.010 | 0.519±0.021 | 0.496±0.044 |
| | 336 | **0.487±0.004** | **0.439±0.003** | 0.491±0.003 | 0.445±0.006 | 0.560±0.009 | 0.543±0.018 | 0.602±0.013 | 0.647±0.038 |
| | 720 | **0.571±0.004** | **0.572±0.003** | 0.587±0.002 | 0.592±0.006 | 0.643±0.017 | 0.680±0.026 | 0.717±0.010 | 0.875±0.006 |
| ExchangeRate | 96 | **0.169±0.001** | **0.054±0.001** | 0.180±0.001 | 0.061±0.001 | 0.184±0.006 | 0.062±0.003 | 0.330±0.039 | 0.190±0.054 |
| | 168 | 0.220±0.005 | 0.092±0.002 | **0.214±0.002** | **0.086±0.002** | 0.223±0.005 | 0.091±0.007 | 0.439±0.069 | 0.320±0.090 |
| | 336 | **0.303±0.000** | **0.165±0.002** | 0.306±0.000 | 0.170±0.001 | 0.313±0.007 | 0.175±0.006 | 0.544±0.067 | 0.507±0.139 |
| | 720 | 0.437±0.007 | 0.338±0.012 | **0.435±0.020** | **0.329±0.022** | 0.610±0.072 | 0.589±0.118 | 0.642±0.088 | 0.648±0.173 |
| Traffic | 96 | 0.344±0.001 | 0.393±0.001 | 0.323±0.001 | 0.389±0.001 | **0.322±0.005** | **0.375±0.006** | 0.354±0.004 | 0.405±0.004 |
| | 168 | 0.348±0.002 | 0.403±0.001 | **0.327±0.001** | 0.404±0.001 | 0.333±0.002 | **0.397±0.002** | 0.356±0.006 | 0.416±0.010 |
| | 336 | 0.360±0.002 | 0.426±0.002 | **0.341±0.000** | 0.429±0.000 | 0.348±0.003 | **0.424±0.002** | 0.371±0.005 | 0.443±0.005 |
| | 720 | **0.377±0.000** | **0.454±0.002** | 0.385±0.001 | 0.472±0.001 | 0.379±0.002 | 0.470±0.003 | 0.390±0.006 | 0.477±0.006 |

# E  Full Results And Discussions

## E.1  Experiment On Synthetic Data

To fully demonstrate the effectiveness of our method on signals with varying non-stationary frequencies, we generated a synthetic multidimensional time-series dataset using composite sinusoidal signals [32] to verify the effectivenss of FAN on evolving non-stationary time series. Each dimension is composed of $i$ superimposed sinusoidal signals with linearly changing amplitude, generated as $\mathcal{X}_t^{(i)} = \sum_{j=1}^{i} a_t \sin \frac{2\pi}{T_j} t, \quad i = 1, \ldots, D.$ where $a_t$ is the signal amplitude, $T_i$ are the periodicities, for example, daily periodicity in hour $T_i = 24$, and $t$ is the current time step.

In the synthetic dataset, each synthetic signal is a combination of multiple sinusoidal signals with linear changes and fixed periods, and varies in the training, validation, and test sets. In Table 7, we list the settings of these signals, every synthetic signal is a composition of these signals, e.g., Syn-5 contains Sig1-5, Syn-9 contains Sig1-9, the generation code can also be found in our code repository2.

Table 7: Synthetic signal settings, the amplitude changes linearly in train/val/test sets.

| Signal | Sig1 | Sig2 | Sig3 | Sig4 | Sig5 | Sig6 | Sig7 | Sig8 | Sig9 |
|---|---|---|---|---|---|---|---|---|---|
| Periodicity | 12 | 16 | 24 | 36 | 48 | 60 | 72 | 84 | 96 |
| Amplitude Change | (0,1,2,4) | (1,3,5,6) | (3,4,6,8) | (1,2,4,5) | (1,3,5,6) | (1,3,5,6) | (1,3,5,6) | (1,3,5,6) | (1,3,5,6) |

We conduct a 720-step multivariate forecasting experiment on synthetic data using DLinear as the backbone model and compared with other reversible normalization methods. Results are shown in Table 8. We observe averaged improvements ranging from 19.75% to 47.04%. As the number of composite frequencies increases (from Syn-5 to Syn-9), the prediction difficulty escalates. Previous normalization methods failed to make further improvements as probably because they can not address seasonality patterns shift. Conversely, our model's enhancements steadily increase. The MAE/MSE improvements ranges from 19.75% to 32.45% and from 23.76% to 41.75%. This underscores the effectiveness of FAN in handling intricate seasonality patterns.

Table 8: Forecasting errors under the multivariate setting. The bold values indicate best performance.

| Methods | FAN | | SAN | | Dish-TS | | RevIN | | Improvement | |
|---|---|---|---|---|---|---|---|---|---|---|
| Metrics | MAE | MSE | MAE | MSE | MAE | MSE | MAE | MSE | MAE | MSE |
| Syn-5 | **0.252** | **0.138** | 0.314 | 0.181 | 0.321 | 0.207 | 0.372 | 0.284 | 19.75% | 23.76% |
| Syn-6 | **0.224** | **0.113** | 0.291 | 0.163 | 0.287 | 0.177 | 0.326 | 0.227 | 21.95% | 30.67% |
| Syn-7 | **0.235** | **0.117** | 0.341 | 0.206 | 0.296 | 0.187 | 0.321 | 0.223 | 31.09% | 43.20% |
| Syn-8 | **0.279** | **0.152** | 0.413 | 0.287 | 0.391 | 0.315 | 0.428 | 0.371 | 32.45% | 47.04% |
| Syn-9 | **0.329** | **0.212** | 0.461 | 0.364 | 0.441 | 0.394 | 0.475 | 0.448 | 25.40% | 41.75% |

We compare the performance of different normalization methods in Fig. 12. Beyond performance, FAN also surpasses other models in performance on synthetic datasets, requiring only three epochs to achieve a smaller test loss compared to ten epochs required by other normalizations. This demonstrates that varying periodic signals indeed affect the predictive performance of models and the non-stationarity that normalization methods must counteract, with our model proving effective in handling these challenges.

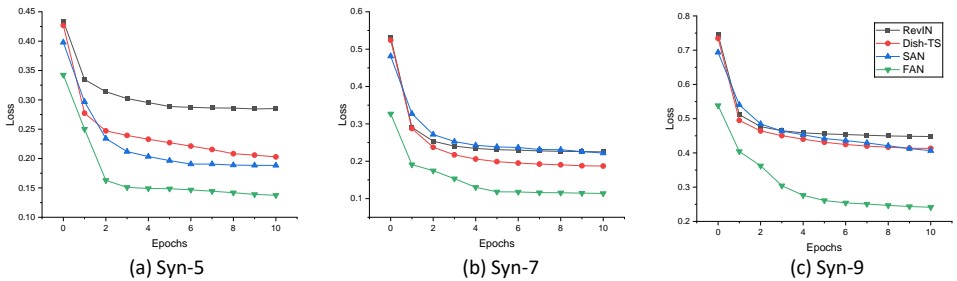

Figure 12: Performance comparison with different normalization baselines, where FAN clearly outperforms others in this varying frequencies condition.

## E.2 Full results of ETT benchmarks

We present the full results of ETT benchmarks in Table 9. In the entire ETT benchmarks, FAN demonstrates improvements over the original models in 114 out of 128 metrics. Specifically, our model exhibits average enhancements of, 18.43%, 31.64%, and 12.04% for FEDformer, Informer, and SCINet respectively. It's worth noting that despite FEDformer's utilization of Fourier transform for seasonality analysis, it still struggles with handling changing seasonality patterns, thus leaving

room for our 31.64% improvement. However, FAN fails to enhance the DLinear backbone in the ETTm1 dataset. We attribute this to the inherently higher stationarity of the ETTm1 dataset and its fewer trend changes (Table 1), which may not align well with the DLinear model which is based on the moving average.

Table 9: Full results on ETT benchmarks. The bold values indicate best performance.

| Methods Metrics | DLinear MAE | MSE | +FAN MAE | MSE | FEDformer MAE | MSE | +FAN MAE | MSE | Informer MAE | MSE | +FAN MAE | MSE | SCINet MAE | MSE | +FAN MAE | MSE |
|---|---|---|---|---|---|---|---|---|---|---|---|---|---|---|---|---|
| **ETTh1** 96 | **0.424** | 0.368 | 0.426 | **0.362** | 0.520 | 0.485 | **0.444** | **0.378** | 0.598 | 0.646 | **0.434** | **0.367** | 0.461 | 0.409 | **0.427** | **0.362** |
| 168 | **0.449** | 0.398 | 0.452 | **0.393** | 0.549 | 0.523 | **0.468** | **0.406** | 0.694 | 0.863 | **0.465** | **0.407** | 0.518 | 0.489 | **0.454** | **0.395** |
| 336 | 0.485 | **0.448** | **0.484** | 0.435 | 0.576 | 0.570 | **0.493** | **0.443** | 0.738 | 0.950 | **0.507** | **0.467** | 0.574 | 0.582 | **0.487** | **0.439** |
| 720 | **0.561** | **0.558** | 0.572 | 0.574 | 0.687 | 0.781 | **0.559** | **0.546** | 0.823 | 1.106 | **0.602** | **0.617** | 0.645 | 0.707 | **0.572** | **0.575** |
| **ETTh2** 96 | **0.237** | 0.110 | 0.238 | 0.112 | 0.312 | 0.176 | **0.263** | **0.126** | 0.298 | 0.160 | **0.256** | **0.124** | 0.264 | 0.128 | **0.239** | **0.112** |
| 168 | 0.254 | 0.127 | **0.253** | **0.129** | 0.339 | 0.199 | **0.275** | **0.140** | 0.331 | 0.191 | **0.269** | **0.138** | 0.292 | 0.156 | **0.255** | **0.130** |
| 336 | 0.271 | 0.138 | **0.267** | **0.142** | 0.334 | 0.194 | **0.294** | **0.157** | 0.347 | 0.208 | **0.300** | **0.162** | 0.305 | 0.167 | **0.269** | **0.142** |
| 720 | 0.316 | 0.179 | **0.281** | **0.158** | 0.360 | 0.238 | **0.304** | **0.174** | 0.413 | 0.291 | **0.378** | **0.256** | 0.339 | 0.201 | **0.284** | **0.159** |
| **ETTm1** 96 | **0.380** | **0.310** | 0.394 | 0.334 | 0.481 | 0.443 | **0.396** | **0.335** | 0.514 | 0.520 | **0.389** | **0.322** | 0.421 | 0.355 | **0.394** | **0.333** |
| 168 | **0.408** | **0.354** | 0.416 | 0.364 | 0.510 | 0.475 | **0.424** | **0.371** | 0.563 | 0.600 | **0.417** | **0.362** | 0.446 | 0.399 | **0.415** | **0.363** |
| 336 | **0.446** | **0.416** | 0.456 | 0.423 | 0.544 | 0.530 | **0.461** | **0.425** | 0.612 | 0.690 | **0.462** | **0.425** | 0.489 | 0.464 | **0.456** | **0.423** |
| 720 | **0.488** | **0.471** | 0.493 | 0.476 | 0.595 | 0.617 | **0.507** | **0.482** | 0.697 | 0.849 | **0.506** | **0.483** | 0.553 | 0.563 | **0.495** | **0.477** |
| **ETTm2** 96 | 0.203 | 0.080 | **0.198** | **0.078** | 0.208 | 0.082 | **0.194** | **0.074** | 0.226 | 0.091 | **0.198** | **0.077** | 0.206 | 0.079 | **0.198** | **0.078** |
| 168 | 0.220 | 0.093 | **0.219** | **0.093** | 0.249 | 0.116 | **0.220** | **0.093** | 0.251 | 0.112 | **0.219** | **0.092** | 0.226 | 0.094 | **0.218** | **0.093** |
| 336 | 0.245 | 0.114 | **0.241** | **0.113** | 0.282 | 0.143 | **0.272** | **0.131** | 0.283 | 0.140 | **0.245** | **0.114** | 0.262 | 0.122 | **0.241** | **0.113** |
| 720 | 0.270 | 0.142 | **0.264** | **0.139** | 0.308 | 0.174 | **0.275** | **0.145** | 0.347 | 0.212 | **0.287** | **0.154** | 0.297 | 0.153 | **0.264** | **0.139** |

### E.3 FAN for Fourier-based Bakcbones

To demonstrate the effectiveness of our method in extracting non-stationary seasonal patterns, we select two other models based on the Fourier transform and perform additional experiments, including: (1) TimesNet [41], which generates 2D variations of time series data using Fourier Transform; (2) Koopa [27], which employs Fourier Transform for dynamic time series modeling based on koopman theory. It is important to note that although both models extract the top $k$ signals, their main frequency selection is based on the average dimensions of the training set rather than the input-specific, which may limit their models' ability to handle varying dimensions and inputs. Furthermore, they internally use RevIN [17] in their model implementation, for a fair comparison, we remove this part or replace it with FAN, we present the experiment results in Table 10.

As in Table 10, Even with state-of-the-art models based on Fourier-transform, our model still demonstrates performance improvements across almost all datasets. Specifically, for long inputs, our model consistently shows performance enhancements. In particular, on the Exchange dataset, our model achieves a maximum improvement of 84.85% in TimesNet and 26.10% in Koopa. We believe the significant improvement in TimesNet is due to its lack of handling seasonal non-stationarity compared to Koopa. However, Although Koopa explicitly handle non-stationarity, we still observe improvements in Koopa. For the ETTm2, Electricity, Traffic, and Weather datasets, our model shows stable MSE performance large improvements for short-term 96 steps and long-term 720 steps inputs by 4.71%/6.09%, 1.30%/8.57%, -1.96%/2.57%, and -1.78%/2.43%, respectively. This improvement is likely due to our model's approach of learning directly from the changes in primary frequency components and our instance-wise and dimension-specific frequency analysis.

### E.4 Full Results of Baselines Comparison

In Table 11, we provide the detailed experimental results of the comparison between FAN and state-of-the-art normalization methods for non-stationary time series normalization: RevIN [17], Dish-TS [10], and SAN [28].

The table clearly shows that FAN outperforms existing approaches in most cases, particularly in the ExchangeRate dataset, our long-term 720-step prediction significantly outperforms other baselines, highlighting the importance of handling seasonal non-stationary information in long-term predictions. However, we fail to make further improvement on Weather and ETTh2 dataset. Considering that in these two datasets, the seasonal variation is very small (as in Appendix B.1), the limited seasonal non-stationary information might have led to the inability to further optimization.

Table 10: Mulltivariate long-term forecasting for Fourier-transform based backbones. The bold letter indicates the best result.

| Models | | TimesNet | | +FAN | | Improvements | | Koopa | | +FAN | | Improvements | |
| Datasets | | MAE | MSE | MAE | MSE | MAE | MSE | MAE | MSE | MAE | MSE | MAE | MSE |
|---|---|---|---|---|---|---|---|---|---|---|---|---|---|
| ETTh1 | 96 | 0.519 | 0.482 | **0.428** | **0.362** | 17.53% | 25.00% | 0.442 | 0.382 | **0.428** | **0.364** | 3.20% | 4.72% |
| | 168 | 0.588 | 0.586 | **0.461** | **0.403** | 21.64% | 31.29% | 0.470 | 0.418 | **0.455** | **0.396** | 3.31% | 5.38% |
| | 336 | 0.694 | 0.838 | **0.507** | **0.467** | 26.87% | 44.25% | 0.498 | 0.471 | **0.489** | **0.442** | 1.78% | 6.01% |
| | 720 | 0.771 | 0.995 | **0.591** | **0.588** | 23.37% | 40.94% | 0.582 | 0.621 | **0.576** | **0.579** | 0.99% | 6.79% |
| ETTh2 | 96 | 0.354 | 0.228 | **0.240** | **0.112** | 32.26% | 50.81% | 0.248 | 0.121 | **0.241** | **0.114** | 2.76% | 5.72% |
| | 168 | 0.364 | 0.238 | **0.275** | **0.144** | 24.68% | 39.58% | 0.262 | 0.137 | **0.255** | **0.131** | 2.72% | 4.94% |
| | 336 | 0.400 | 0.284 | **0.343** | **0.218** | 14.27% | 23.36% | 0.275 | 0.150 | **0.269** | **0.143** | 2.03% | 4.10% |
| | 720 | 0.509 | 0.519 | **0.415** | **0.309** | 18.49% | 40.57% | 0.294 | 0.176 | **0.283** | **0.159** | 3.61% | 9.75% |
| ETTm1 | 96 | 0.491 | 0.471 | **0.388** | **0.326** | 20.89% | 30.63% | 0.407 | 0.353 | **0.395** | **0.336** | 2.84% | 4.71% |
| | 168 | 0.528 | 0.530 | **0.411** | **0.353** | 22.25% | 33.39% | 0.430 | 0.384 | **0.415** | **0.363** | 3.51% | 5.44% |
| | 336 | 0.592 | 0.595 | **0.454** | **0.414** | 23.27% | 30.39% | 0.474 | 0.451 | **0.465** | **0.434** | 1.93% | 3.76% |
| | 720 | 0.710 | 0.865 | **0.498** | **0.474** | 29.79% | 45.22% | 0.519 | 0.516 | **0.502** | **0.485** | 3.29% | 6.09% |
| ETTm2 | 96 | 0.237 | 0.096 | **0.194** | **0.075** | 18.16% | 22.35% | 0.207 | 0.083 | **0.203** | **0.079** | 1.94% | 4.66% |
| | 168 | 0.283 | 0.144 | **0.215** | **0.090** | 23.87% | 37.73% | 0.226 | 0.099 | **0.219** | **0.093** | 2.89% | 5.76% |
| | 336 | 0.313 | 0.175 | **0.245** | **0.115** | 21.92% | 34.06% | 0.249 | 0.122 | **0.242** | **0.114** | 2.84% | 6.64% |
| | 720 | 0.344 | 0.211 | **0.283** | **0.155** | 17.90% | 26.52% | 0.280 | 0.158 | **0.268** | **0.140** | 4.18% | 11.46% |
| Electricity | 96 | 0.359 | 0.256 | **0.248** | **0.154** | 31.05% | 39.90% | 0.278 | 0.181 | **0.264** | **0.179** | 5.08% | 1.30% |
| | 168 | 0.370 | 0.264 | **0.253** | **0.157** | 31.67% | 40.60% | 0.283 | 0.189 | **0.272** | **0.181** | 3.80% | 4.15% |
| | 336 | 0.382 | 0.272 | **0.268** | **0.165** | 29.85% | 39.31% | 0.304 | 0.209 | **0.294** | **0.196** | 3.28% | 6.30% |
| | 720 | 0.400 | 0.292 | **0.288** | **0.180** | 27.79% | 38.53% | 0.341 | 0.251 | **0.331** | **0.230** | 2.93% | 8.57% |
| Exchange | 96 | 0.507 | 0.380 | **0.174** | **0.058** | 65.64% | 84.85% | **0.169** | 0.055 | 0.170 | **0.055** | -0.27% | 0.50% |
| | 168 | 0.596 | 0.523 | **0.223** | **0.095** | 62.57% | 81.91% | **0.216** | **0.088** | 0.217 | 0.089 | -0.21% | -1.19% |
| | 336 | 0.703 | 0.723 | **0.307** | **0.169** | 56.30% | 76.61% | 0.317 | 0.182 | **0.299** | **0.162** | 5.68% | 10.82% |
| | 720 | 0.749 | 0.810 | **0.442** | **0.341** | 40.96% | 57.84% | 0.514 | 0.414 | **0.418** | **0.306** | 18.70% | 26.10% |
| Traffic | 96 | 0.350 | 0.418 | **0.306** | **0.349** | 12.65% | 16.47% | 0.352 | **0.420** | 0.351 | 0.428 | 0.37% | -1.96% |
| | 168 | 0.353 | 0.434 | **0.315** | **0.375** | 10.84% | 13.54% | 0.359 | 0.442 | **0.352** | **0.436** | 1.82% | 1.42% |
| | 336 | 0.356 | 0.446 | **0.322** | **0.383** | 9.55% | 14.16% | 0.371 | 0.467 | **0.363** | **0.458** | 2.22% | 1.84% |
| | 720 | 0.372 | 0.475 | **0.344** | **0.416** | 7.62% | 12.42% | 0.392 | 0.499 | **0.383** | **0.486** | 2.48% | 2.57% |
| Weather | 96 | 0.464 | 0.436 | **0.223** | **0.174** | 51.92% | 60.15% | **0.198** | **0.169** | 0.214 | 0.172 | -8.00% | -1.78% |
| | 168 | 0.528 | 0.543 | **0.263** | **0.214** | 50.22% | 60.58% | **0.233** | **0.211** | 0.255 | 0.213 | -9.41% | -0.75% |
| | 336 | 0.589 | 0.663 | **0.338** | **0.308** | 42.67% | 53.60% | **0.284** | **0.286** | 0.299 | 0.275 | -5.51% | 3.79% |
| | 720 | 0.703 | 0.950 | **0.426** | **0.426** | 39.42% | 55.19% | 0.344 | 0.349 | **0.324** | **0.340** | 5.64% | 2.43% |

# F  Limitations

Though FAN shows promising performance, there are still some limitations. First, we removed a significant amount of non-stationary trend and seasonal information. While this is effective in most baselines, it may lead to an over-stationary issue, causing a decline in the backbone's performance. Second, we largely select the frequency counts $K$ in a search-based manner or based on dataset priors. This approach may lead to incorrect $K$ value selection, resulting in an under-stationary issue or more severe over-stationary issue. Additionally, our non-stationary pattern extraction is based on the Fourier transform, and a finite number of Fourier signals cannot represent all periodic signals, which may hinder our ability to handle some waveforms, e.g. square waves. Therefore, future work can focus on more effective instance-specific $K$ value selection, strict dynamic control of non-stationarity elimination, and other methods for extracting non-stationary waveforms.

Table 11: Forecasting errors under the multivariate setting. The bold values indicate best performance.

| Models | | DLinear | | | | FEDFormer | | | | Informer | | | | SCINet | | |
|---|---|---|---|---|---|---|---|---|---|---|---|---|---|---|---|---|---|
| Methods | | FAN | SAN | Dish-TS | RevIN | FAN | SAN | Dish-TS | RevIN | FAN | SAN | Dish-TS | RevIN | FAN | SAN | Dish-TS | RevIN |
| ETTh1 96 | MAE | **0.426** | 0.432 | 0.433 | 0.428 | **0.444** | 0.491 | 0.509 | 0.523 | **0.434** | 0.498 | 0.556 | 0.521 | **0.427** | 0.431 | 0.438 | 0.438 |
| | MSE | **0.362** | 0.370 | 0.375 | 0.375 | **0.378** | 0.453 | 0.477 | 0.495 | **0.367** | 0.466 | 0.549 | 0.517 | **0.362** | 0.370 | 0.382 | 0.380 |
| ETTh1 168 | MAE | **0.452** | 0.460 | 0.454 | 0.464 | **0.468** | 0.511 | 0.531 | 0.554 | **0.465** | 0.514 | 0.601 | 0.539 | **0.454** | 0.459 | 0.476 | 0.470 |
| | MSE | **0.393** | 0.404 | 0.405 | 0.416 | **0.406** | 0.486 | 0.505 | 0.548 | **0.407** | 0.485 | 0.642 | 0.531 | **0.395** | 0.404 | 0.430 | 0.425 |
| ETTh1 336 | MAE | **0.484** | 0.504 | 0.505 | 0.501 | **0.493** | 0.551 | 0.575 | 0.568 | **0.507** | 0.627 | 0.662 | 0.642 | **0.487** | 0.502 | 0.539 | 0.490 |
| | MSE | **0.435** | 0.463 | 0.475 | 0.476 | **0.443** | 0.549 | 0.583 | 0.575 | **0.467** | 0.702 | 0.753 | 0.735 | **0.439** | 0.461 | 0.522 | 0.462 |
| ETTh1 720 | MAE | **0.572** | 0.584 | 0.590 | 0.598 | **0.559** | 0.603 | 0.646 | 0.653 | **0.602** | 0.689 | 0.739 | 0.763 | **0.572** | 0.579 | 0.604 | 0.584 |
| | MSE | **0.574** | 0.579 | 0.603 | 0.641 | **0.546** | 0.633 | 0.695 | 0.747 | **0.617** | 0.845 | 0.914 | 0.968 | **0.575** | 0.580 | 0.622 | 0.620 |
| ETTh2 96 | MAE | **0.234** | 0.237 | 0.237 | 0.239 | 0.263 | **0.264** | 0.291 | 0.285 | **0.256** | 0.272 | 0.330 | 0.309 | 0.239 | **0.238** | 0.265 | 0.241 |
| | MSE | **0.108** | 0.112 | 0.111 | 0.117 | 0.126 | **0.132** | 0.163 | 0.155 | **0.124** | 0.138 | 0.196 | 0.178 | **0.112** | 0.113 | 0.132 | 0.115 |
| ETTh2 168 | MAE | **0.251** | 0.252 | 0.255 | 0.255 | 0.275 | **0.270** | 0.312 | 0.296 | **0.269** | 0.296 | 0.361 | 0.317 | 0.255 | **0.252** | 0.281 | 0.263 |
| | MSE | **0.126** | 0.128 | 0.129 | 0.135 | **0.140** | 0.141 | 0.185 | 0.166 | **0.138** | 0.159 | 0.234 | 0.189 | 0.130 | **0.127** | 0.152 | 0.140 |
| ETTh2 336 | MAE | **0.263** | 0.264 | 0.269 | 0.273 | 0.294 | **0.303** | 0.361 | 0.325 | **0.300** | 0.310 | 0.375 | 0.334 | 0.269 | **0.263** | 0.297 | 0.275 |
| | MSE | **0.132** | 0.137 | 0.138 | 0.152 | 0.157 | **0.174** | 0.247 | 0.198 | **0.162** | 0.176 | 0.254 | 0.205 | 0.142 | **0.136** | 0.165 | 0.151 |
| ETTh2 720 | MAE | **0.281** | 0.286 | 0.288 | 0.303 | 0.304 | **0.297** | 0.374 | 0.335 | 0.378 | 0.416 | 0.436 | **0.354** | 0.284 | 0.304 | 0.321 | 0.305 |
| | MSE | **0.158** | 0.159 | 0.165 | 0.193 | 0.174 | **0.174** | 0.274 | 0.214 | 0.256 | 0.332 | 0.350 | **0.223** | **0.159** | 0.179 | 0.190 | 0.190 |
| ETTm1 96 | MAE | 0.394 | 0.386 | 0.407 | **0.383** | **0.396** | 0.418 | 0.467 | 0.473 | **0.389** | 0.401 | 0.457 | 0.446 | 0.394 | **0.389** | 0.415 | 0.436 |
| | MSE | 0.334 | **0.311** | 0.356 | 0.317 | **0.335** | 0.371 | 0.443 | 0.460 | **0.322** | 0.330 | 0.445 | 0.420 | 0.333 | **0.321** | 0.357 | 0.423 |
| ETTm1 168 | MAE | **0.416** | 0.416 | 0.421 | 0.435 | **0.424** | 0.439 | 0.502 | 0.506 | **0.417** | 0.443 | 0.496 | 0.470 | **0.415** | 0.422 | 0.442 | 0.454 |
| | MSE | 0.364 | **0.354** | 0.373 | 0.390 | **0.371** | 0.387 | 0.493 | 0.501 | **0.362** | 0.393 | 0.496 | 0.457 | **0.363** | 0.367 | 0.414 | 0.430 |
| ETTm1 336 | MAE | **0.456** | 0.458 | 0.459 | 0.480 | **0.461** | 0.473 | 0.532 | 0.537 | **0.462** | 0.492 | 0.536 | 0.524 | 0.456 | **0.454** | 0.481 | 0.490 |
| | MSE | 0.423 | **0.415** | 0.433 | 0.463 | **0.425** | 0.437 | 0.536 | 0.550 | **0.425** | 0.460 | 0.552 | 0.525 | 0.423 | **0.415** | 0.467 | 0.486 |
| ETTm1 720 | MAE | **0.493** | 0.497 | 0.501 | 0.530 | **0.507** | 0.521 | 0.579 | 0.571 | **0.506** | 0.545 | 0.608 | 0.597 | **0.495** | 0.498 | 0.515 | 0.525 |
| | MSE | 0.476 | **0.468** | 0.492 | 0.534 | **0.482** | 0.507 | 0.608 | 0.594 | **0.483** | 0.552 | 0.659 | 0.678 | 0.477 | **0.473** | 0.510 | 0.536 |
| ETTm2 96 | MAE | 0.198 | **0.197** | 0.207 | 0.202 | **0.194** | 0.195 | 0.218 | 0.210 | **0.198** | 0.201 | 0.238 | 0.210 | 0.198 | **0.197** | 0.206 | 0.197 |
| | MSE | 0.078 | **0.077** | 0.082 | 0.080 | **0.074** | 0.077 | 0.095 | 0.084 | **0.077** | 0.079 | 0.105 | 0.086 | 0.078 | 0.077 | 0.083 | **0.077** |
| ETTm2 168 | MAE | 0.219 | **0.217** | 0.222 | 0.224 | 0.220 | **0.219** | 0.231 | 0.227 | **0.219** | 0.221 | 0.261 | 0.235 | 0.218 | **0.217** | 0.227 | 0.220 |
| | MSE | 0.093 | **0.092** | 0.094 | 0.097 | **0.093** | 0.093 | 0.103 | 0.099 | **0.092** | 0.094 | 0.133 | 0.105 | 0.093 | **0.093** | 0.099 | 0.094 |
| ETTm2 336 | MAE | **0.241** | 0.242 | 0.246 | 0.250 | 0.272 | **0.244** | 0.266 | 0.270 | **0.245** | 0.249 | 0.302 | 0.275 | 0.241 | **0.240** | 0.258 | 0.250 |
| | MSE | **0.113** | 0.114 | 0.114 | 0.121 | 0.131 | **0.116** | 0.139 | 0.137 | **0.114** | 0.120 | 0.169 | 0.142 | 0.113 | **0.113** | 0.126 | 0.122 |
| ETTm2 720 | MAE | **0.264** | 0.268 | 0.274 | 0.277 | 0.275 | **0.267** | 0.293 | 0.287 | 0.287 | 0.293 | 0.336 | 0.314 | 0.264 | **0.262** | 0.303 | 0.277 |
| | MSE | **0.139** | 0.142 | 0.144 | 0.155 | 0.145 | **0.140** | 0.164 | 0.162 | 0.154 | 0.162 | 0.207 | 0.186 | 0.139 | **0.137** | 0.181 | 0.155 |
| Electricity 96 | MAE | **0.266** | 0.284 | 0.278 | 0.273 | **0.243** | 0.258 | 0.277 | 0.270 | **0.248** | 0.280 | 0.303 | 0.275 | 0.258 | 0.269 | 0.289 | **0.251** |
| | MSE | **0.181** | 0.189 | 0.189 | 0.198 | **0.148** | 0.154 | 0.170 | 0.168 | **0.152** | 0.171 | 0.195 | 0.172 | 0.165 | 0.164 | 0.185 | **0.151** |
| Electricity 168 | MAE | **0.267** | 0.281 | 0.273 | 0.267 | **0.251** | 0.270 | 0.285 | 0.277 | **0.252** | 0.288 | 0.320 | 0.279 | 0.258 | 0.272 | 0.301 | **0.254** |
| | MSE | **0.177** | 0.183 | 0.181 | 0.184 | **0.154** | 0.164 | 0.176 | 0.173 | **0.155** | 0.178 | 0.211 | 0.177 | 0.163 | 0.168 | 0.200 | **0.155** |
| Electricity 336 | MAE | **0.288** | 0.301 | 0.296 | 0.289 | **0.272** | 0.282 | 0.294 | 0.289 | **0.272** | 0.312 | 0.335 | 0.299 | 0.278 | 0.287 | 0.312 | **0.266** |
| | MSE | **0.191** | 0.198 | 0.197 | 0.201 | **0.167** | 0.172 | 0.181 | 0.180 | **0.167** | 0.197 | 0.222 | 0.192 | 0.175 | 0.177 | 0.207 | **0.162** |
| Electricity 720 | MAE | **0.322** | 0.333 | 0.340 | 0.329 | 0.300 | **0.299** | 0.312 | 0.308 | **0.304** | 0.332 | 0.357 | 0.326 | 0.312 | 0.307 | 0.336 | **0.296** |
| | MSE | **0.224** | 0.231 | 0.239 | 0.244 | 0.189 | **0.184** | 0.198 | 0.200 | **0.194** | 0.217 | 0.249 | 0.218 | 0.204 | 0.193 | 0.237 | **0.188** |
| Exchange 96 | MAE | 0.167 | 0.166 | 0.202 | **0.164** | 0.186 | **0.166** | 0.273 | 0.207 | 0.182 | **0.168** | 0.278 | 0.223 | 0.169 | **0.166** | 0.220 | 0.170 |
| | MSE | 0.053 | 0.054 | 0.070 | **0.053** | 0.062 | **0.054** | 0.138 | 0.082 | 0.061 | **0.055** | 0.183 | 0.096 | **0.054** | 0.054 | 0.087 | 0.057 |
| Exchange 168 | MAE | 0.217 | **0.213** | 0.277 | 0.216 | 0.222 | **0.216** | 0.366 | 0.267 | 0.239 | **0.238** | 0.364 | 0.295 | 0.220 | **0.213** | 0.303 | 0.218 |
| | MSE | 0.088 | **0.087** | 0.127 | 0.088 | 0.090 | **0.089** | 0.221 | 0.130 | 0.105 | 0.110 | 0.279 | 0.157 | 0.092 | **0.087** | 0.186 | 0.089 |
| Exchange 336 | MAE | **0.297** | 0.304 | 0.332 | 0.312 | 0.336 | **0.301** | 0.415 | 0.364 | **0.329** | 0.406 | 0.566 | 0.375 | **0.303** | 0.305 | 0.439 | 0.314 |
| | MSE | **0.162** | 0.171 | 0.190 | 0.178 | 0.198 | **0.166** | 0.274 | 0.231 | **0.184** | 0.305 | 0.603 | 0.252 | **0.165** | 0.171 | 0.318 | 0.183 |
| Exchange 720 | MAE | **0.406** | 0.466 | 0.628 | 0.526 | **0.436** | 0.498 | 0.680 | 0.647 | **0.431** | 0.599 | 0.730 | 0.503 | 0.437 | 0.474 | 0.583 | 0.496 |
| | MSE | **0.292** | 0.375 | 0.674 | 0.440 | **0.329** | 0.461 | 0.699 | 0.625 | **0.322** | 0.591 | 0.822 | 0.448 | 0.338 | 0.386 | 0.534 | 0.403 |
| Traffic 96 | MAE | **0.334** | 0.374 | 0.403 | 0.556 | 0.326 | **0.315** | 0.344 | 0.336 | **0.314** | 0.323 | 0.351 | 0.372 | **0.340** | 0.358 | 0.391 | 0.371 |
| | MSE | **0.403** | 0.443 | 0.513 | 0.738 | 0.371 | **0.350** | 0.389 | 0.377 | **0.364** | 0.365 | 0.415 | 0.455 | **0.393** | 0.409 | 0.458 | 0.434 |
| Traffic 168 | MAE | **0.334** | 0.517 | 0.585 | 0.598 | 0.336 | **0.324** | 0.360 | 0.351 | **0.319** | 0.340 | 0.355 | 0.506 | **0.346** | 0.348 | 0.392 | 0.356 |
| | MSE | **0.414** | 0.654 | 0.796 | 0.803 | 0.391 | **0.376** | 0.421 | 0.410 | **0.383** | 0.400 | 0.423 | 0.746 | **0.403** | 0.412 | 0.468 | 0.418 |
| Traffic 336 | MAE | **0.346** | 0.371 | 0.394 | 0.379 | 0.348 | **0.336** | 0.372 | 0.370 | **0.333** | 0.403 | 0.376 | 0.636 | 0.357 | **0.356** | 0.403 | 0.366 |
| | MSE | **0.437** | 0.463 | 0.511 | 0.520 | 0.414 | **0.406** | 0.454 | 0.446 | **0.406** | 0.518 | 0.459 | 1.048 | 0.426 | 0.437 | 0.498 | 0.444 |
| Traffic 720 | MAE | **0.372** | 0.395 | 0.420 | 0.403 | 0.372 | **0.364** | 0.387 | 0.381 | **0.397** | 0.563 | 0.402 | 0.786 | 0.377 | **0.375** | 0.423 | 0.382 |
| | MSE | **0.472** | 0.497 | 0.541 | 0.548 | 0.454 | **0.449** | 0.469 | 0.463 | **0.482** | 0.778 | 0.489 | 1.327 | 0.454 | 0.465 | 0.533 | 0.473 |
| Weather 96 | MAE | **0.214** | 0.228 | 0.247 | 0.216 | 0.252 | 0.255 | 0.406 | **0.217** | 0.217 | 0.219 | 0.251 | **0.203** | 0.215 | 0.219 | 0.234 | **0.196** |
| | MSE | **0.173** | 0.175 | 0.190 | 0.195 | **0.187** | 0.198 | 0.380 | 0.191 | 0.172 | **0.170** | 0.190 | 0.173 | 0.170 | **0.164** | 0.175 | 0.164 |
| Weather 168 | MAE | 0.254 | 0.258 | 0.285 | **0.242** | 0.304 | 0.269 | 0.438 | **0.269** | 0.247 | 0.253 | 0.303 | 0.248 | 0.253 | 0.257 | 0.270 | **0.232** |
| | MSE | 0.210 | **0.206** | 0.226 | 0.231 | 0.240 | **0.217** | 0.450 | 0.255 | 0.208 | **0.206** | 0.255 | 0.228 | 0.206 | **0.203** | 0.213 | 0.207 |
| Weather 336 | MAE | 0.297 | 0.312 | 0.342 | **0.290** | 0.366 | 0.419 | 0.556 | **0.299** | 0.315 | 0.316 | 0.376 | **0.306** | 0.299 | 0.309 | 0.314 | **0.288** |
| | MSE | **0.274** | 0.277 | 0.293 | 0.301 | 0.321 | **0.393** | 0.672 | 0.310 | 0.287 | **0.279** | 0.364 | 0.314 | **0.268** | 0.269 | 0.275 | 0.285 |
| Weather 720 | MAE | 0.345 | 0.358 | 0.400 | **0.327** | 0.441 | 0.469 | 0.625 | **0.445** | 0.368 | 0.379 | 0.454 | **0.350** | 0.340 | 0.355 | 0.355 | **0.356** |
| | MSE | 0.339 | **0.338** | 0.366 | 0.359 | **0.432** | 0.479 | 0.746 | 0.486 | **0.360** | 0.368 | 0.479 | 0.386 | **0.322** | 0.331 | 0.336 | 0.348 |

