# OpenReview forum: "Frequency Adaptive Normalization For Non-stationary Time Series Forecasting"
_NeurIPS.cc/2024/Conference — NeurIPS 2024 poster_

### Official Review · Reviewer_Cudr · 2024-07-01

**Soundness:** 3
**Presentation:** 3
**Contribution:** 3
**Rating:** 7
**Confidence:** 4

**Summary:**

This paper introduces FAN, a novel instance normalization technique designed to address both dynamic trends and seasonal patterns. FAN is a model-agnostic method that can be integrated with various predictive models. It significantly enhances performance, achieving average MSE improvements of 7.76% to 37.90%.

**Strengths:**

- Non-stationary time series forecasting has long been a highly challenging problem. Despite extensive research, there remains significant value in further exploration.
- This paper is well-written, precise in expression, and comprehensive in content. It offers a new perspective on mitigating non-stationarity issues in non-stationary time series forecasting.
- The experiments in this paper are relatively comprehensive.

**Weaknesses:**

- There are some shortcomings in the experimental descriptions. For example, the calculation methods for Trend Variation and Seasonality Variation are not clearly explained, and the ADF test values are provided only after normalization, lacking a direct comparison with the values before normalization.
- The selection of the hyperparameter K plays a crucial role in the effectiveness of the model.

**Questions:**

See the weakness.

**Limitations:**

Limitations are in the appendix.

---

> ### Author Rebuttal · Authors · 2024-08-06
>
> We appreciate all your suggestions and hope that the following answers have clarified your questions.
>
> **Q1: lacking experimental descriptions, e.g. dataset metrics Trend  and Seasonality Variation.**
>
> Thank you for your valuable advice. Due to page limitations, we have briefly discussed the calculation methods for the data in the main text (line 185-187, 248). Based on your suggestion, additional details will be provided in the appendix to improve clarity and reproducibility. Furthermore, we have also included a Jupyter notebook demonstrating the calculation of selected K and other metrics, e.g. Trend/Seasonal Variation, ADF test values, available on  [anonymous repo](https://github.com/icannotnamemyself/FAN/blob/main/notebooks/metrics.ipynb). Speciffically, tread/seasonal variations are calculated as follows:
>
> **Trend Variation**:, given a timeseries dataset $\mathcal{X} \in \mathbb{R}^{N\times D}$, we first chronologically split it into $\mathcal{X}^{\text{train}}$, $\mathcal{X}^{\text{val}}$, and $\mathcal{X}^{\text{test}}$, representing the training, validation, and testing datasets, respectively. The trend variations are then computed as follows:
> $$
>  \text{Trend Variation} = \left|\frac{ \operatorname{Mean}_N(\mathcal{X}^{\text{train}}) - \operatorname{Mean}_N(\mathcal{X}^{\text{val}, \text{test}})}{\operatorname{Mean}_N(\mathcal{X}^{\text{train}})}\right|
> $$
> where the subscripts indicate the dimension of mean, $| \cdot |$ denotes the absolute value operation, and $\mathcal{X}^{\text{val}, \text{test}}$ represents the concatenation of the validation and test sets. Note that, to obtain relative results across different datasets, the trend variation is normalized by dividing by the mean of the training dataset. We fetch the first dimension to be the value in main content Table 1.
>
> **Sesonal Variation**:Given the inputs, $X  \in \mathbb{R} ^ {N_i \times L \times D}$, where $N_i$ is the number of inputs, we first obtain the FFT results of all inputs, denoted as $Z \in \mathbb{C} ^ {N_i \times L \times D}$. Then, we calculate the variance across different inputs and normalize this variance by dividing it by the mean of each input, computed as:
>
> $$
> \text{Seasonal Variation} = \frac{\text{Var}_{N_i}[\text{Amp}(Z)]}{\text{Mean}_L(X)}
> $$
>
> where the subscripts indicate the dimension of the operation. We sum the results across all channels for the value in the main text, Table 1.
>
>
>
>
> **Q2: lacking the ADF test values before normalization.**
>
> Thanks for the suggestion, we will include the original values in Table 1 in the latest version. Since we perform the ADF test after normalization at the instance level and RevIN's can bot handle internal non-stationarity within the input, **the values before normalization and the values after applying RevIN are too similar to distinguish** in Fig 4(a), as shown in the [anonymous notebook](https://github.com/icannotnamemyself/FAN/blob/main/notebooks/metrics.ipynb). Therefore, we only plot the RevIN values for a clearer comparison. However, we will include the original values in Table 1 following your advice.
>
>
>
> **Q3: The selection of the hyperparameter K.**
>
> Although we introduced a hyperparameter K, its selection is relatively straightforward based on the distribution in the frequency domain of the dataset (Fig 7); moreover, to avoid selecting the parameter K, we provide a heuristic formula selecting frequencies greater than 10% of the maximum amplitude (line 207) which is applied across our experiments. We further illustrate the effectiveness of this selection rule in **attached PDF Fig 1**.
>
> We hope our responses have adequately addressed your concerns, and we are eager to provide further insights into our study.

---

> ### Comment · Reviewer_Cudr · 2024-08-11
>
> I appreciate the authors' detailed responses, which address my concerns and I will raise the rating of the paper to 7.

---

> > ### Author Response · Authors · 2024-08-12
> > **Thanks for your positive feedback**
> >
> > Dear Reviewer Cudr,
> >
> > we sincerely value your feedback and the constructive suggestions you've provided for enhancing our paper. If you have any further questions or concerns, please feel free to let us know.
> >
> > Authors

---

### Official Review · Reviewer_MHdq · 2024-07-04

**Soundness:** 3
**Presentation:** 3
**Contribution:** 3
**Rating:** 6
**Confidence:** 5

**Summary:**

The paper introduces Frequency Adaptive Normalization (FAN) to improve time series forecasting by addressing non-stationary data with evolving trends and seasonal patterns. Unlike reversible instance normalization, which handles trends but not seasonal patterns, FAN employs the Fourier transform to identify and model predominant frequency components. This model-agnostic approach, applied to four forecasting models, shows significant performance improvements across eight benchmark datasets.

**Strengths:**

1. The paper is well-written and structured, with a well-motivated idea and promising experimental results.


2. The code is available for checking and reproducing the results.

**Weaknesses:**

1. I noticed a significant disparity between the results reported in Table 2 and those reported by SAN, despite using the same model configurations. This performance gap is unexpected and warrants further investigation.


2. A key question is whether FAN's capabilities fully encompass those of SAN. Specifically, can FAN handle changes in statistical properties, such as mean and variation, that characterize seasonal and trend patterns? A more detailed comparison between FAN and previous relevant methods would be beneficial.


3. I noticed that FAN consumes 0.24 million parameters, which is significantly more than FITS [1], an entirely frequency-based benchmark model. It would be helpful to clarify why such a large number of parameters is necessary to address non-stationarity.


4. Previous studies [1][2] have shown that purely using frequency-domain representation can achieve accurate future variation estimation with simple, lightweight models. The proposed method, however, relies on a backbone model in addition to FAN to handle specific parts of the evolution. Does this combination offer an advantage over methods based solely on frequency components?


5. Furthermore, can FAN enhance frequency-based methods in a similar manner to DLinear and FEDformer?

[1] Xu, Zhijian, Ailing Zeng, and Qiang Xu. "FITS: Modeling time series with $10 k $ parameters." arXiv preprint arXiv:2307.03756 (2023).

[2] Yi, Kun, et al. "Frequency-domain MLPs are more effective learners in time series forecasting." Advances in Neural Information Processing Systems 36 (2024).

**Questions:**

Please address the weak points I raised above.

**Limitations:**

There is no discussion on the limitations or social impacts of this paper.

---

> ### Author Rebuttal · Authors · 2024-08-06
>
> We appreciate your constructive suggestions as they can aid in enhancing our work. We hope that responses below have addressed your concerns:
>
> **Q1: performance gap in Tab 2 and reported by SAN.**
>
> Thanks for this question. We have rechecked the results, papers, and codes to identify the reasons for the discrepancy between our results and those of SAN. The main reason is the **different data splitting methods used.** In results of SAN, the split ratio is 6:2:2 for the ETT dataset and 7:1:2 for the other datasets, while in our experiments, we used a split ratio of 7:2:1 for all datasets (as stated on line 185).  We opted for this setup to unify the experimental setup, thereby increasing the reliability of the results.
>
>
> **Q2: whether FAN's capabilities fully encompass those of SAN, can FAN handle changes in statistical properties?**
>
> FAN can encompass SAN if we ignore variance as both FAN and SAN can be adjusted to achieve a constant zero mean in distribution. We have provided a theoretical analysis of FAN's effect on temporal statistical properties in **Sec C.3**. Based on the conclusions from the analysis in Sec C.3, FAN can handle statistical changes, as the mean will be zero and the variance variations will be largely reduced.
>
> **Q3: A more detailed comparison between FAN and previous relevant methods.**
>
>  A comprehensive comparison is already provided in **Sec 4.3** covering aspects such as prediction performance and showcases, stationarity after normalization, model efficiency, etc. Additional results for the synthetic dataset can also be found in **Sec E.1**.
>
> To address your concern, we list the table below to summarize key metrics and give as many additional details as possible:
>
> |Method|techique|trend|seasonality|space(M)|training iter time(ms)|performance|training strategy|modification on input statistics|
> |---|------|-----|-----|--------|-------|----|-------|---|
> |FAN|Fourier-based|✅|✅|✅0.249|✅0.32|✅|✅end-to-end|$\mu=0$, $\sigma<<\sigma_{raw}$|
> |SAN(2024)|statistics-based|✅|❌|❌0.351|❌0.32x2|⛔|❌two-stages|$\mu=0$, $\sigma=1$|
> |DishTS(2023)|statistics-based|✅|❌|⛔0.307|0.43⛔|❌|✅end-to-end|$\mu=0$, $\sigma=1$|
> 1. ✅: better ❌: worse ⛔: average, SAN and DishTS are two sotas in  recent years.
> 2.  The actual training iteration time should be doubled to justify the average iteration time of SAN which is trained on two-stages.
>
> We hope this has met your expectation, and we would appreciate it if you could provide any other comparisons that might help improve our work.
>
> **Q4: why FAN consumes 0.24 million parameters, which is significant compared to FITS.**
>
> Thanks for the insightful question and for bringing up the issue of parameters. In Fig 4b, the parameters are actually consist of DLinear (0.140 M), FAN (0.109M). Why FAN consumes 109k which is large compared to FITS is as follows:
>
> Firstly, while FITS reduces the number of parameters in the input and output layers by assuming a constant dominant frequency and using upsampling, **FAN aims to model varying dominant frequencies**; hence, we cannot reduce parameters by modeling only a fixed subset of dominant  frequencies, as FITS does.
>
> Secondly, FAN has tunable parameters and **reducing the parameters does not significantly impact performance**, but **we chose a rather high performance with acceptable parameters  across our paper ([64,128,128])**, as in the table below:
>
> |Method|MSE|Parameter|
> |-|-|-|
> |FAN-[4,8,8]|0.1421|7.2k|
> |FAN-[8,16,16]|0.1413|14.6k|
> |FAN-[16,32,32]|0.1406|28.9k|
> |FAN-[32,64,64]|0.1395|58.1k|
> |FAN-[64,128,128]|**0.1390**|109k|
> |FAN-[128,256,256]|0.1392|255k|
> |FITS|0.1423|**5.5k**|
> |FreTS|0.1391|3516k|
> - the experiment is conducted on ETTm2 with $H=720$ without backbone.
>
> As in table, after reducing the parameters to 7.2k, FAN can still achieve better performance to FITS (5.5k), possiblely due to  our instance-wise selection of dominant frequencies (**Sec 4.5**).
>
> **Q5 : Does FAN's combination of a backbone offer an advantage over methods based solely on frequency components?**
>
> In **Fig 9**, it can be seen that the changes in the dominant frequency are relatively small compared to the changes in the residual frequencies; furthermore, our experiments in **Tab 6** show that model more parameters to predict the dominant frequency did not yield better results, which may be due to the relatively small and stable variations in the dominant frequency.**Therefore, incorporating the backbone indeed provides some advantages, as we use a robust model to predict the relatively small changes in the dominant frequency and a rather complex model to predict the larger variations in the residual frequencies.**
>
> Furthermore, although we did find that even without the backbone, FAN already demonstrates good predictive capabilities by just forecasting varying dominant frequencies. **This combination with the backbone indeed further improves performance and we have provided an ablation study in the main content Sec 4.5 Tab 4 to show the performance variations.**
>
> **Q6: can FAN enhance frequency-based methods in a similar manner to DLinear and FEDformer?**
>
> Thanks for this question. FEDformer is itself a frequency-based method. And we specifically chose this backbone to compare FAN's performance on frequency-based method. Futhermore, we discussed the results and analysis of frequency-based methods, FEDformer/TimesNet/Koopa on line 50, line 92-98, lines 293-304, and **Sec E.3**. Also, we provide results of FITS and FreTS at **attached PDF Tab 2**.
>
> As in the result, FAN can also enhance FITS/FreTS even they are predicting based only frequency components. This is possibly due to our instance-wise selection of dominant frequencies, treating them as the primary non-stationary component and predicting the residual frequencies separately.
>
>  We hope our explanations have sufficiently answered your questions, and we value the opportunity to explain our work further.

---

> > ### Author Response · Authors · 2024-08-12
> >
> > Dear Reviewer MHdq,
> >
> >   We noticed that you have increased our score to 6, and we sincerely appreciate your positive feedback. We highly value all your efforts during the rebuttal phase. If you have any further questions or concerns, please do not hesitate to let us know.
> >
> > Authors

---

### Official Review · Reviewer_3SM3 · 2024-07-12

**Soundness:** 4
**Presentation:** 4
**Contribution:** 3
**Rating:** 8
**Confidence:** 5

**Summary:**

This paper proposes a new instance normalization solution called frequency adaptive normalization (FAN) to address non-stationary data in time series forecasting. This paper extends instance normalization to handle both dynamic trend and seasonal patterns by employing Fourier transform to identify predominant frequency components. This paper introduces a simple MLP model to predict the discrepancy of frequency components between inputs and outputs. Several experiments are conducted to prove the effectiveness of the proposed method.

**Strengths:**

1. this paper is easy to understand. The writing is good.

2. this paper is well-motivated. Since the current normalization techniques are not specifically focused on periodic patterns, the authors try to use frequency techniques to enhance them.

3. the experiments are well-sounded. Several experiments are conducted.

4. the overall framework is simple but effective, which is good for time series forecasting against distribution shifts.

**Weaknesses:**

1. Missing related work. Several frequency-focused or periodicity-focused works can be discussed [1-3].

2. Most adopted backbones seem to be earlier works of time series forecasting. Adding more backbones such as PatchTST can make the experiments better.

[1] DEPTS: Deep expansion learning for periodic time series forecasting. In ICLR.

[2] Frequency-domain MLPs are more effective learners in time series forecasting. In NeurIPS.

[3] Deep Frequency Derivative Learning for Non-stationary Time Series Forecasting. In IJCAI.

**Questions:**

Can more related works be discussed?
Can more results be provided?

**Limitations:**

See weakness.

---

> ### Author Rebuttal · Authors · 2024-08-06
>
> We are grateful for your positive feedback on our work, hope that the answers provided below have resolved your inquiries:
>
> **Q1: missing related work. e.g. DEPTS (ICLR 2022), FreTS (NIPS 2024),  DERITS(IJCAI 2024).**
>
> Thank you for bringing these works to our attention! These studies indeed help us enrich the related work section. Specifically, DEPTS models periodic states as hidden states and uses an expansion module to model the relationship between cycles and future prediction steps; FreTS uses MLPs to model both the channel and temporal dimensions in the frequency domain; DERITS employs reverse transformation to map the time series to the whole spectrum through K branches FDT and uses iFDT to generate output.
>
> We will include these works to provide a comprehensive review of frequency-focused or periodicity-focused works in the camera-ready version.
>
>
> **Q2: more results, e.g. PatchTST.**
>
> Thanks for the advice. Here is the result of using PatchTST as backbone where FAN can indeed enhance PatchTST on most metrics. More results can be found at **attached PDF Tab 2**.
>
>
> | Methods      |     | PatchTST  |           | +FAN      |           |
> | ------------ | --- | --------- | --------- | --------- | --------- |
> | Metric       |     | MAE       | MSE       | MAE       | MSE       |
> | ETTm2        | 96  | 0.202     | 0.079     | **0.199** | **0.078** |
> |              | 168 | 0.225     | 0.097     | **0.219** | **0.093** |
> |              | 336 | **0.239** | **0.112** | 0.242     | 0.114     |
> |              | 720 | 0.272     | 0.142     | **0.268** | **0.141** |
> | Electricity  | 96  | 0.263     | 0.180     | **0.254** | **0.153** |
> |              | 168 | 0.263     | 0.176     | **0.255** | **0.158** |
> |              | 336 | 0.282     | 0.189     | **0.275** | **0.169** |
> |              | 720 | 0.319     | 0.220     | **0.300** | **0.189** |
> | ExchangeRate | 96  | 0.189     | 0.063     | **0.172** | **0.056** |
> |              | 168 | 0.237     | 0.102     | **0.225** | **0.097** |
> |              | 336 | 0.333     | 0.198     | **0.293** | **0.160** |
> |              | 720 | 0.470     | 0.355     | **0.428** | **0.324** |
> | Traffic      | 96  | 0.323     | 0.384     | **0.314** | **0.374** |
> |              | 168 | **0.330** | **0.406** | 0.334     | 0.414     |
> |              | 336 | **0.338** | **0.427** | 0.340     | 0.430     |
> |              | 720 | 0.378     | 0.460     | **0.373** | **0.454** |
> | Weather      | 96  | 0.222     | 0.173     | **0.220** | **0.170** |
> |              | 168 | 0.257     | 0.210     | **0.251** | **0.209** |
> |              | 336 | 0.305     | 0.283     | **0.301** | **0.278** |
> |              | 720 | 0.363     | 0.351     | **0.350** | **0.344** |
>
>
>  Based on your advice, we will try to include these results in the latest version.
>
> We hope our explanations have sufficiently answered your questions, and we value the opportunity to explain our work further.

---

> > ### Comment · Reviewer_3SM3 · 2024-08-12
> >
> > Since the authors have addressed all my questions, I have raised my score.

---

> ### Author Response · Authors · 2024-08-12
> **Thanks for your positive feedback**
>
> Dear Reviewer 3SM3,
>
> We are grateful for your positive evaluation of our work. We greatly appreciate the time and effort you have invested in reviewing our work. If you have any further questions, concerns, or suggestions for improvement, please feel free to reach out to us.
>
> Authors

---

### Official Review · Reviewer_LAap · 2024-07-12

**Soundness:** 3
**Presentation:** 2
**Contribution:** 2
**Rating:** 3
**Confidence:** 3

**Summary:**

This paper presents FAN - frequency adaptive normalization - as an alternative approach to de-trending seasonality in non-stationary data through Fourier transform decomposition. The method relies on dynamically identifying K instance-wise predominant frequency components; the evolution of these components is then modeled using a simple MLP approach rather than assuming they remain unchanged. The authors employ FAN for eight benchmark forecasting problems using four different ML backbones and assess the performance with and without FAN. The authors provide the code through a public GitHub repository.

**Strengths:**

The paper addresses the problem of evolving seasonality in non-stationary data through utilizing instance-wise frequency components, rather than global frequency analysis, coupled with an MLP approach to capture non-linear relationships in the evolution of the predominant non-stationary components. The main contribution appears to be a dynamic selection of the K relevant frequencies, as opposed to previous methods that assume a fixed frequency set across inputs or select frequencies randomly, as well as modeling the frequency components’ evolution using an MLP approach rather than assuming they remain constant or through predicting the statistics.

**Weaknesses:**

While the contributions appear novel and the paper presents both theoretical considerations and applied results, the robustness of the methodology is not well established by the authors. The authors state that the selection of K is done based on inspection of the data as “the average maximum amplitude within 10% of the training set” and is shown to vary between 2 and 30 for the benchmark datasets. However, the ablation study (which is presented only in the Appendix and showcases only a subset of the datasets) does not properly support the selection method -  the selected K for the presented datasets were 3, 5, 2, and 30 while the sensitivity analysis only utilizes K = 4, 8, 12, and 24. It would have been more pertinent to do dataset-specific analysis as the hyperparameter is dataset-specific. Furthermore, there appears to be a prediction length dependency that is not properly investigated. In terms of establishing the performance of FAN, the authors present comparisons between the performance of different backbone models with and without FAN as the main result and relegate comparison to other normalization methods to averages over all prediction lengths in the main paper. While the premise of improving prediction performance through normalization is indeed established through the former comparison, that is not the scope of the paper. Furthermore, we are unable to reproduce the percentage MAE and MSE improvements stated for FAN in section 4.2, especially those used to claim improved performance with prediction length.

The paper showcases an overall logical organization and contains most necessary sections for properly presenting the work (relegating the limitations section to the Appendix is a questionable choice). However, there are several places where grammar and phrasing reduce clarity and make readability difficult. A few examples: the description of the selection of the hyperparameter K as “the average maximum amplitude within 10% of the training set” is not clear; FAN is presented as a normalization *method* that can be combined with any ML model, however, in Section E1, FAN is referred to as a *model*; the y-axis in Fig 4b is nonsensical and overestimates the difference between models; Fig 10 would be clearer if it showed difference between input and output frequencies rather than overlaying the two. Also, the figures throughout the paper and Appendix would benefit from increased font size (most are barely legible without significant zoom).  Further, there are a few statements throughout the paper that indicate overconfidence in the results, or confusion in interpreting them. For example, Fig 4c is used to support FAN’s improved performance with increasing input length compared to other models. While MSE is indeed lower for FAN for longer input lengths, all models show the same trend in decreased MSE with increased input length. Similarly, Fig 12 is stated to demonstrate the higher convergence speed of FAN compared to other normalization methods. Again, the metric (loss as opposed to MSE in Fig 4c) is indeed lower for FAN, but the compared-to methods seem to plateau at earlier epochs than FAN, indicating the opposite of the claim of which methods converge faster.

While the proposed normalization method does present some novelty that can be built upon by the authors and others, the significance of the contribution is difficult to assess due to the aforementioned limitations in proven robustness of methodology. The results would also benefit from statistical analysis to properly elucidate the improved performance of the methods - simply indicating lower errors while not investigating if the error reduction is significant compared to other methods and using models without FAN does not properly support FAN’s performance.

**Questions:**

1. What are the assumptions for correlation between input dimensions?
2. How come you select a data normalization method (z-score) that does not handle non-stationary series when your data is inherently non-stationary? How does this affect your results?
3. How have you calculated the MAE and MSE improvements, specifically for Tab 2? We are not able to recreate the stated percentages, and cannot support the statement about improved performance with increased prediction length.
4. What is the comparison between on line 266?
5. In Section C4 it is stated that "the range of the distribution mean has decreased to 8" - decreased from what? How is this seen in Fig 10?

**Limitations:**

The authors do not present limitations in the main paper but have opted to relegate this section to the Appendix. While we understand that the page restrictions for the paper puts restraints on the presented content, adjustments should be made to ensure important information is presented in the main paper. The content of the limitation section is also not satisfactory and, in fact, enforces the aforementioned concerns about the contributions of the paper. Namely, the authors discuss their proposed methodology for selection of the hyperparameter K, which is presented as one of the novel contributions of their works, and state that their approach “may lead to incorrect K value selection”. It would have been advisory to focus the current study on elucidating the robustness of the proposed methodology, rather than simply presenting results of one implementation.

---

> ### Author Rebuttal · Authors · 2024-08-06
>
> We thank Reviewer LAap for such a  comprehensive review of our work. We hope we have addressed all your concerns as follows:
>
> **Q1:  the ablation study and sensitivity analysis are provided only in the appendix and only utilizes K = 4, 8, 12, 24, which should be dataset-specific and not all dataset results have been provided.**
>
> In the main content, we do provide an ablation study in Sec 4.5 and sensitivity analysis in Sec 4.4, and K ranges from 1 to 32, containing all selected K across datasets.
>
> Futhermore, as shown in table below, due to the large differences in spectrum distribution across datasets (Fig 7), it is hard to make cross-comparisons when using different ratios in experiments. Therefore, we chose a consistent range for K to better present the results.
>
> |ratios|0.1|0.2|0.3|
> |------|---|---|---|
> |Exchange|3|2|1|
> |Weather|3|2|1|
> |Electricity|18|3|1|
> |ETTm2|7|5|2|
> |Traffic|49|30|16|
>
>
> However, to address your concern, we provide results on all datasets at **attched PDF Fig 1**, with dataset-specific K ranging from 1 to 49. This  will be included in the latest version.
>
>
> **Q2: Reproducibility.**
>
> we have double checked our code, we are sure you can reproduce the results in our paper. To assist your review, we add some model checkpoints and training logs for you to test performance and compare training process in the [anonymous repo](https://github.com/icannotnamemyself/FAN/tree/main/results/runs/DLinear/ExchangeRate). The docs is updated accordingly.
>
>
> **Q3: several places where grammar and words reduce clarity and  readability.**
>
> Thank you for carefully reviewing our paper. We will recheck the grammar/wording and make appropriate changes based on your suggestions.
>
> **Q4: statements indicate overconfidence:  (1) Fig 12 use "convergence speed". (2) Fig 4c where all methods' MSE/MAE are decreasing.**
>
> Thank you for bringing up these issues. (1) We acknowledge that the word "convergence speed" may lack rigor; we will update the term to "better convergence" in the latest version. (2) The conclusion derivation in Fig 4c is consistent with previous approaches, e.g.  Table 5 in DishTS, Fig 4 in SAN, Fig 7 in RevIN. The mentioned issues also exist in these works. We will consider changing to relative improvement rather than absolute metric values in the latest version.
>
> Nevertheless, we are certain that except these issues, the remaining conclusions of this paper are correctly stated based on objective results.
>
>
> **Q5: What are the assumptions for correlation between input dimensions?**
>
> We follow a channel independence assumption for input dimensions (mentioned in line 129).
>
> **Q6: Why use z-score and its effect on results.**
>
> As in lines 192, and in previous works, e.g. SAN, z-score can "scale the dataset to the same scale, facilitating results readibility". Furthermore, it can help model training, e.g. the loss might be too small without scaling transformation. To evaluate its effect, we conduct experiments on Traffic/ETTh1 with $H=720$ and use DLinear as backbone, the MSE results are:
>
> Traffic:
> |Method|z-score|w/o z-score|
> |-|-|-|
> |FAN|**0.472**|**0.00126**|
> |w/o FAN|0.532|0.00249|
> |IMP| 11.27%|49.40%|
>
> ETTh1:
>
> |Method|z-score|w/o z-score|
> |-|-|-|
> |FAN|**0.158**|**15.686**|
> |w/o FAN|0.179|19.940|
> |IMP|11.73%|21.33%|
>
> As shown in tables above, without z-score, the metrics are difficult to read/present. **Futhermore, the improvements made by FAN  without z-score are even more pronounced**. For reviewers to reproduce the result, we update the repo docs accordingly.
>
> **Q7:  How the improvements are calculated? specifically for Tab 2.**
>
> Thanks for the valuable question! The formula of MAE/MSE improvements is $f^*_{D,M} (\frac{MSE_{base}-MSE_{our}}{MSE_{base}})$ except in Tab 2 where both MSE and MAE are presented; in Tab 2, the average MSE and MAE improvement is $\frac{\sum_{D,M} (MSE_{base}+MAE_{base})-\sum_{D,M} (MSE_{our}+MAE_{our})}{\sum_{D,M} (MSE_{our}+MAE_{our})}$, since we combine both MAE/MSE results to give an uniform result. We believe that the former is a standard algorithm for calculating improvements, whereas the latter is not. For clarity, we would use the former formula across our paper in the latest version and separately list the improvements in MSE and MAE results.
> The improvements in line 218 regarding Tab 2 will be updated to 9.87%/7.49%, 18.87%/14.73%, 36.91%/25.20%, 16.26%/13.24%, and 20.05%/18.79% for MSE/MAE, respectively. Note that this improvements are still substantial, which highlights the overall performance of FAN.
> - `*`: $D$ and $M$ denote datasets and models, $f$ is some operations, e.g. max, avg across dataset/model, MAE follows the same procedure.
>
> **Q8: stated percentage improvements regarding the prediction length can not be reproduced.**
>
> Thanks for your careful review! After revising the calculation formula (Q7) for improvements in Tab 2, despite the conclusion regarding prediction length remaining valid for Informer (line 223), it cannot be generalized to all models. Therefore, we will remove this conclusion in the latest version.
>
> **Q9: the comparison on line 266.**
>
> The comparison is MSE compared to second best model SAN. The MSE improvements over SAN, increases from 0.49%(L=48) to 4.37%(L=336). We will add more detail on this to increase clarity.
>
>
> **Q10: Sec C4, what decreased to 8? How is this seen in Fig 10?**
>
> As shown in Fig 10, numbers on the polar axis shows the mean amplitude  values of each frequency across the inputs. So, the amplitude distribution mean range decreased from 80 (SAN), 70 (DishTS), 70 (RevIN) to 8 (FAN).
>
> We hope our responses have addressed your concerns, and we appreciate the opportunity to clarify our work.

---

> > ### Comment · Reviewer_LAap · 2024-08-11
> >
> > Thank you for answering my questions. I still have concerns regarding the scientific rigor, quality of the methods, results, and conclusions.
> >
> > Q1. Sec 4.5 is indeed an ablation study but not the one I was referring to in my comment. I am, admittedly, not an expert in this field and so there is a possibility that I have misunderstood details of the presented work. However, my understanding is that the contributions in the presented work are (1) a dynamic selection of K and (2) using MLP to model the freq components evolution rather than assuming they remain constant. As such, the manuscript should focus on the effects of these aspects compared to alternative normalization methods to properly elucidate the benefits of the proposed methodology. I find that this is not the case. Instead, the manuscript focuses on the performance of predictive models with and without FAN. While it is indeed of interest to establish that normalization with FAN does improve performance, this does not put FAN in relation to alternative normalization methods.
> > Thank you for providing the additional figure and tables. I suggest looking them over once more to ensure that values in Tab 1 are correct. “The green background corresponds to the selected K across our experiments” seems to be wrong for ETTh1, ETTh2, ETTm1, and ETTm2.
> >
> > Q2. My comment on reproducibility was not on the full study but on the percentage MAE and MSE improvements for FAN in section 4.2, the same as for my question 3. You address my question below (Q8) and seemingly admit to there being an error in the calculation.
> >
> > Q4. Regarding the convergence speed: “Better convergence” is a weak statement that is still incorrect. Convergence refers to the plateauing of loss, not the absolute value of the loss. In fact, for Syn-7 and Syn-9, the FAN appears to result in slower convergence than the alternative normalization methods.
> > Regarding the input length: It is not the statement that FAN shows a reduction in MSE with increased input lengths that is the issue (the statement appears true based on Fig 4c), rather the phrasing “compared to other models”. This phrasing is incorrect (all models show this behavior) and misleads the reader. Is the MSE significantly lower for FAN at longer inputs compared to other models? If yes, then this is a significant result that should be pointed out. If not, then this is not a significant result and should not be made out to be “Notably”.This goes back to my original comment about backing your results up with statistical tests to prove significance in your presented numbers.
> >
> > Q6. I am not questioning scaling the data, this is indeed necessary for the model training. I am questioning the choice of scaling method. I see that z-score is used also in the SAN and RevIN papers. Did you assess the distribution of the datasets prior to opting for z-score or was the choice based on the use in the aforementioned papers? Did you consider dynamic scaling alternatives that would have been more suitable for non-stationary data?
> >
> > Q7. Thank you for the clarification. MAE and MSE do not provide the same information and should hence not be combined.
> >
> > Q10. Thank you for the clarification. I highly suggest adding a more descriptive caption to the figure to help the reader understand the plots. As stated in my original review, I would also suggest displaying the difference between input and output rather than overlaying the two or, alternatively, splitting them up and presenting them in separate plots.

---

> > > ### Author Response · Authors · 2024-08-12
> > >
> > > We appreciate your time in reviewing our paper and giving us the opportunity to clarify our work further. We understand that there may be some misunderstanding, as you are not specialized in this field; however, we hope the following response will fully address any misunderstandings you may have about our work.
> > >
> > >
> > > Q1: Our contribution has been clearly illustrated in main content **Sec 1**, and you have acknowledged these contributions in the strength of the original comments:  (1) instance-wise select top K main components, not "dynamic selection of K" (2) modeling varying main frequency through MLP. We believe there are some misunderstandings regarding how to demonstrate the validity of these contributions. However,  **we do not focus solely on the performance of predictive models with or without FAN**, to address your issue, here is a list:
> > >
> > >
> > > 1. **contribution 1**: In Sec 4.5 (Fig 6, Tab 5),  we analyze the distribution of the primary frequencies to illustrate the necessity of selecting the instance-wise primary frequencies, not by merely listing the results;
> > >
> > > 2. **contribution 2**: In Sec 4.4, 4.5 Tab 4, Sec B.3 and D.2, we explain that a simple MLP might achieve better performance by analyzing relative variation of main and residual frequency components across datasets.
> > >
> > > 3. **comparison with alternative normalization RevIN, DishTS, SAN**: In Sec 4.3, we provide a comprehensive comparison regarding stationarity test after normalization, model efficiency, etc.
> > >
> > > 4. **full data analysis**: In Sec B, we conducted a comprehensive analysis of frequency distribution, selection distribution, and other relevant factors to further validate the necessity of Contribution 1 and Contribution 2.
> > >
> > > 5. **Theoretical analysis**: In Sec C, we discuss the effect of FAN on Fourier and temporal distribution, theoretically.
> > >
> > >
> > > Hence, **a significant portion of our paper is based on model comparison/data analysis/theoretical results   rather than merely on results with or without FAN**.
> > >
> > >
> > >
> > > We admit some typos of ETT datasets in material Tab 1 and thanks for pointing out, below is the updated version, others remain the same:
> > >
> > > | |ETTh1 | ETTh2 | ETTm1 | ETTm2 |
> > > |-------|-------|-------|-------|--|
> > > | 0     | 49    | 49    | 49    | 49 |
> > > | 0.05  | 16    | 11    | 11    | 7  |
> > > | 0.1   | 4     | 3     | 11    | 5  |
> > > | 0.2   | 3     | 3     | 3     | 2  |
> > > | 0.3   | 3     | 1     | 3     | 2  |
> > > | 0.4   | 3     | 1     | 3     | 1  |
> > > | 0.5   | 2     | 1     | 2     | 1  |
> > >
> > >
> > > We sincerely apologize for making this mistake, and are willing to provide any code/notebooks to prove the reproducibility of our work.
> > >
> > >
> > > Q2/Q7: We combined the results, not miscalculated, and this does not affect the overall conclusion of the paper. As in the rebuttal Q7: "the improvements in line 218 regarding Tab 2 will be updated to 9.87%/7.49%, 18.87%/14.73%, 36.91%/25.20%, 16.26%/13.24%, and 20.05%/18.79% for MSE/MAE, respectively", and the improvements are still substantial.
> > >
> > >
> > > Q4: Fig 4c on line 264 is just a description of the results. We do not claim this as a generalized conclusion, as whether larger input length will contain more varying seasonal patterns is dataset-specific. We will remove notably as it might be too strong according to your advice.
> > >
> > >
> > > Q6: **Z-score normalization is used mainly to scale data for better readability and presentation**, and we adopt z-score following prior work, e.g. SAN. **And this is actually a convention in the time series domain, e.g. DLinear, FEDformer, SCINet**. Furthermore, FAN is a reversible instance normalization method for handling non-stationarity. **Addressing non-stationarity during the preprocessing stage would hinder the demonstration of our model's effectiveness** in processing non-stationarity.
> > >
> > >
> > > Note that the reversible normalization were introduced specifically to address issues caused by dynamic normalization. For example, sliding window normalization might mitigate distributions shift, but they remove too much information, making it difficult for the backbone model to predict the removed content.  we recommend you  refer to a pioneering work in this area, RevIN, for more details. Therefore, we do not adopt these dynamic normalization methods.
> > >
> > >
> > >
> > > Q10: We will try our best to clarify the captions and adjust the tables/figures based on your valuable advice.
> > >
> > > We hope above responses have addressed all your concerns. We noticed that Reviewer MHdq and Reviewer Cudr increased the score from 5 to 6 and 6 to 7 respectively, and you decreased the score from 4 to 3. If we may ask, could you please clarify the reason for lowering the score from 4 to 3? This will help us clarify any misunderstandings and improve our work further.
> > >
> > > We sincerely look forward to your reply.

---

### Author Rebuttal · Authors · 2024-08-07

Dear Reviewers, ACs and the SAC:

We thank you all for the review and valuable comments. We'll clarify them in the final version to address all relevant questions and suggestions.

To address the common concerns regarding our selection of K (Reviewer LAap, Reviewer Cudr) and our model effectiveness on more backbones (Reviewer 3SM3, Reviewer MHdq), we provide explanations and additional results in the **attached PDF**.


We are grateful for your helpful advice, as it can support the advancement of our work. We hope the those responses have satisfied your queries.

---

### Comment · Area_Chair_Q2dU · 2024-08-12
**Please Engage Promptly in Author-Reviewer Discussions**

Dear Reviewers,

As we approach the end of the Author-Reviewer Discussion phase, please take the time to read all reviews and author responses thoroughly and post your response. This will ensure there is ample opportunity for meaningful back-and-forth discussion, allowing the authors to clarify and strengthen their work.

Your timely participation is key to the success of this phase. Thank you for your continued dedication and effort!

-AC

---

### Decision · Program_Chairs · 2024-09-25

**Decision:**

Accept (poster)

**Comment:**

This paper proposes a novel normalization method named Frequency Adaptive Normalization (FAN), to handle non-stationary time series with evolving trend and seasonal patterns. The authors demonstrate the effectiveness of the FAN method combined with existing forecasting models on multiple benchmark datasets, showing significant performance improvements over existing normalization methods.

During the rebuttal period, the primary concerns include selecting and analyzing the hyperparameter K, clarifying the experimental setup, details on evaluation metrics, contribution discussion, and extra analysis with SAN/FreTS/FEDformer/..., etc. The authors provided detailed rebuttals and responses with extra experiments to answer these questions. Following this, three reviewers increase their scores to accept, leaving only one negative score with concerns about scientific rigor and the quality of the results. Based on the whole paper and the rebuttal, I think the authors effectively addressed the main concerns raised in their response. The comprehensive experiments and detailed analysis in the paper, along with the supplementary results provided during the rebuttal, demonstrate the quality of the work. Overall, the strengths of the paper outweigh its weaknesses, which can be revised in the camera-ready version. Therefore, I recommend accepting the paper.